# Exploring Motif-based Heterogeneous Graph Learning for ReDoS Detection

Hong Huang [1 2 3]   Chengyu Yao [4]   Rongchen Li [1 2 3]   Weihao Su [1 2 3]   Chengyao Peng [1 2 3]   Haiming Chen [1 2]
Guiyi He [1 2 3]

## Abstract

Regular expressions (regexes) frequently exhibit super-linear worst-case behavior in regex engines, exposing software to Regex Denial-of-Service (ReDoS) attacks. Detecting such vulnerabilities is challenging, especially for extended features such as lookarounds and backreferences: existing static approaches are efficient but often lack support for extended features, whereas dynamic and hybrid approaches reduce false positives by executing regex matching on real engines, but incur high runtime overhead. To address this trade-off, we propose ReDoS-MotifGNN (RMGNN), a motif-based graph learning framework for ReDoS detection that leverages the low inference latency of graph neural networks (GNNs). RMGNN converts regexes into Heterogeneous Regex Graphs (HRGs) and encodes three ReDoS-related motifs into HRGs to incorporate domain priors, while preserving the syntactic structure and extended features of the input regex. Furthermore, it applies heterogeneous propagation with kernel-guided motif learning to capture multi-scale semantics, which are fused via residual cross-attention for robust prediction. Comprehensive evaluation on four real-world datasets (over 317k regexes) demonstrates that RMGNN outperforms six state-of-the-art baselines in F1-score and achieves an average 244× speedup over the top F1-performing baseline.

## 1. Introduction

As a key tool for pattern matching and searching, regular expressions (regexes) with extended features (e.g., lookarounds, backreferences) have been widely used across diverse fields in computer science such as software engineering, network security, string processing, and databases (Chapman et al., 2017; Bartoli et al., 2016; Davis et al., 2018; 2019; Shen et al., 2018; Michael et al., 2019). However, the application of regexes also brings serious security issues, namely Regex Denial-of-Service (ReDoS) attacks (Bhuiyan et al., 2023; Davis et al., 2018; Staicu & Pradel, 2018). ReDoS exploits the super-linear worst-case behavior (i.e., polynomial or exponential matching time w.r.t. the input length) of regexes, which can lead to catastrophic backtracking in regex engines (Goyvaerts, 2020; Kadlec, 2017), thereby depleting computational resources. Recent statistics (Davis et al., 2018; Michael et al., 2019; Davis, 2020) indicate that up to 10% of regexes in real-world software projects are vulnerable to ReDoS attacks, with incidents such as outages in Stack Overflow (Stack Exchange, 2016) and Cloudflare (The Cloudflare Blog, 2019), highlighting the severity and prevalence of these vulnerabilities. Consequently, early and accurate detection of ReDoS vulnerabilities is essential.

Existing conventional techniques for ReDoS detection can be classified into the following approaches. Static approaches (Kirrage et al., 2013; Weideman et al., 2016; Wüstholz et al., 2017; Parolini & Miné, 2023) are efficient and scalable; however, they usually report many false positives. Further, modeling the ReDoS for extended features, such as backreferences, which are non-regular and whose membership problem is NP-complete (Aho, 1991), still remains a challenge. Dynamic and hybrid approaches (Petsios et al., 2017; Shen et al., 2018; Noller et al., 2018; Liu et al., 2021; Li et al., 2021; McLaughlin et al., 2022; Barlas et al., 2022; Wang et al., 2023) test vulnerabilities in regex engines to achieve higher precision, but typically incur high computational costs. Besides, several learning-based studies leverage machine learning (ML) or large language models (LLMs) for related tasks, including reactive ReDoS detection via resource-usage monitoring (Demoulin et al., 2019), evaluating regexes generated by LLMs (Siddiq et al., 2024a), and PoC generation (Simsek et al., 2025). However, these

[1]Key Laboratory of System Software (Chinese Academy of Sciences), Beijing, China [2]Institute of Software, Chinese Academy of Sciences, Beijing, China [3]University of Chinese Academy of Sciences, Beijing, China [4]China Southern Power Grid Company Limited, Beijing, China. Correspondence to: Haiming Chen <chm@ios.ac.cn>.

*Proceedings of the 43rd International Conference on Machine Learning*, Seoul, South Korea. PMLR 306, 2026. Copyright 2026 by the author(s).

approaches are generally not tailored for the proactive detection of ReDoS vulnerabilities at the regex level.

To address the above problem, in this paper, we propose to use graph neural networks (GNNs) for ReDoS detection, inspired by the success of GNNs in code vulnerability detection. As we know, GNNs can capture control-, data-, and syntax-level dependencies embedded in source code and uncover vulnerability patterns that are difficult for conventional handcrafted analyses to express, owing to their strong structural learning capability. Moreover, GNNs support highly efficient inference with linear time complexity (Wu et al., 2020), stable and low latencies on the order of milliseconds to seconds (Zhou et al., 2021), which makes them particularly attractive for vulnerability detection in latency-sensitive domains such as software supply chains. These properties motivate us to apply GNNs for ReDoS detection. In fact, there has been a growing body of learning-based works leverages GNNs to actively detect code vulnerabilities (Zhou et al., 2019; Şahin et al., 2022; Nguyen et al., 2022; Wen et al., 2023; Chu et al., 2024; Cao et al., 2024). It is well known that regexes can be viewed as a domain-specific programming language for string processing and exhibit structured patterns similar to code.

One key idea behind our method is the design of a motif-enhanced Heterogeneous Regex Graph (HRG). In detail, instead of operating on abstract syntax trees (ASTs) that only encode syntactic hierarchy, we convert each AST into a HRG. The HRG preserves the AST backbone, capturing the structure of extended features, while enriching semantic attributes and dependencies (e.g., counting bounds, lookarounds, and backreferences). To inject ReDoS-related prior knowledge, we further annotate three counting-centered motifs (i.e., key substructures) on top of the HRG at a higher level of abstraction. These motif priors play an auxiliary role in representation learning, as they are designed to capture abstract structural patterns rather than fine-grained handcrafted ones. Each extracted motif instance is assigned a deterministic label and attached to the corresponding HRG nodes/edges as additional attributes, leaving the underlying graph structure unchanged.

Building on HRGs, we propose ReDoS-MotifGNN (RMGNN), a framework for detecting ReDoS vulnerabilities that leverages the stable, low latency of GNN inference architectures. RMGNN adopts a hierarchical representation learning framework that jointly models local relational semantics and global vulnerability structures. It first performs heterogeneous information propagation to encode fine-grained syntactic dependencies into node representations. These embeddings are then abstracted into motif representations through a pooling mechanism, where graph kernel alignment encourages consistency with counting-centered structural priors. Finally, a cross-attention fusion

module dynamically integrates global motif semantics with local node information via a residual connection for accurate ReDoS vulnerability detection. Experiments on four regex datasets show that RMGNN outperforms six state-of-the-art baselines with relative F1 improvements of 6.13%–88.18%, while maintaining up to $2.86\times$–$375\times$ faster inference speed.

To summarize, our contributions are as follows:

- We design a motif-enhanced HRGs representation, which extends ASTs with multi-dimensional attributes and integrates three counting-centered motifs as structural priors.

- We propose RMGNN, a novel framework that employs heterogeneous propagation and a kernel-guided motif pooling mechanism to effectively capture rich structural features.

- We perform comprehensive experiments for evaluating RMGNN, which confirms our method empirically outperforms six state-of-the-art baselines across four real-world regex datasets for ReDoS detection.

## 2. Related Works

Due to space limitations, more detailed related works are provided in Appendix A.

**ReDoS Detections.** Existing conventional ReDoS detection methods are categorized into static, dynamic, and hybrid approaches. Static analysis (Kirrage et al., 2013; Weideman et al., 2016; Wüstholz et al., 2017; Parolini & Miné, 2023) inspects regex structures without execution, so it is efficient but often suffers from high false positives/negatives and provides limited support for extended features (e.g., lookarounds and backreferences). Dynamic and hybrid approaches (Petsios et al., 2017; Shen et al., 2018; Noller et al., 2018; Liu et al., 2021; Li et al., 2021; McLaughlin et al., 2022; Barlas et al., 2022; Wang et al., 2023) validate inputs on real engines to achieve more precise than static analysis, but typically incur significant computational costs and rely heavily on the quality of input generation. Recently, several learning-based studies have explored related directions. RegexClassifier (Lu et al., 2023) explores GNN-based regex analysis with NFA inputs, but its NFA-based formulation cannot directly handle non-regular extended features. Other studies include assessing LLM-generated regexes (Siddiq et al., 2024a), performing reactive ReDoS detection via resource usage monitoring (Demoulin et al., 2019), and generating PoCs (Simsek et al., 2025). However, these methods do not target proactive, regex-level identification of ReDoS vulnerabilities. RMGNN is designed as a scalable detector for proactive vulnerability scanning. It learns regex structures through a novel learning model that supports extended

features, while concrete attack exploitation can be delegated to downstream dynamic/hybrid tools when needed.

**GNNs for Vulnerability Detection.** GNNs are highly effective in capturing structural and semantic code information for vulnerability detection. While early methods reasoned over unified program graphs (Yamaguchi et al., 2014; Allamanis et al., 2018), subsequent works (Schlichtkrull et al., 2018; Hu et al., 2020; Zhou et al., 2019; Wang et al., 2022) highlighted the necessity of modeling multi-relational, heterogeneous data to enhance detection accuracy. Recent advancements have further optimized this paradigm through deeper architectures (Cheng et al., 2021; Cao et al., 2021), statement-level granularity (Hin et al., 2022), and explainability (Wen et al., 2023; Chu et al., 2024; Cao et al., 2024). Moreover, Heterogeneous Graph Transformers (HGTs) (Zhang et al., 2024a; Sun et al., 2025) have recently been adopted to better capture long-range dependencies in complex code structures. Inspired by these advances, we detect ReDoS by modeling regex ASTs as heterogeneous graphs and applying relation-aware GNNs to capture their structural dependencies.

# 3. Preliminaries

**Real-world Regular Expressions.** Real-world regular expressions over a finite alphabet $\Sigma$ unify classical constructs with extended features. Let $\Sigma^*$ be the set of all strings over $\Sigma$, and $\mathbb{N}$ be the set of natural numbers. The formal syntax is defined as:

$$r ::= \quad \varepsilon \mid \varnothing \mid a \mid [\mathrm{C}] \mid r|r \mid rr \mid (r)_i \mid \backslash i \mid r^{\{m,n\}} \mid r^{\{m,\}}$$
$$\mid r^* \mid r^+ \mid r^? \mid r^{*?} \mid r^{+?} \mid r^{??} \mid (?=r) \mid (?!r)$$
$$\mid (?<=r) \mid (?<!r) \mid \hat{} \mid \$ \mid \backslash b \mid \backslash \mathrm{B}$$

where $a \in \Sigma$, $\mathrm{C} \subseteq \Sigma$, and $m, n \in \mathbb{N}$ with $m \le n$. This syntax includes: (i) capturing-group operators $(r)_i$ with their corresponding backreference operators $\backslash i$; (ii) union operators $r|r$; (iii) counting operators, including unbounded ($r^*$, $r^+$, $r^{\{m,\}}$) and bounded forms ($r^{\{m,n\}}$), as well as their lazy forms ($r^{*?}$, $r^{+?}$); (iv) optional operators ($r^?$ and $r^{??}$); and (v) zero-width assertion operators, including lookarounds $(?=r), (?!r), (?<=r), (?<!r)$, together with anchor operators $\hat{}$, $\$$, $\backslash b$, and $\backslash \mathrm{B}$ for boundary matching. A detailed description of these operators is provided in Appendix B.

**Regex Denial of Service (ReDoS).** Regex Denial of Service (ReDoS) refers to an algorithmic-complexity attack in which an adversary exploits structural ambiguities in a regular expression to trigger worst-case backtracking behavior in the underlying regex engine. Formally, a regex $\mathcal{R}$ is said to be *ReDoS-vulnerable* on an engine $\mathcal{M}$ if there exist input strings $\mathcal{W}$ such that the matching cost $\mathrm{Cost}(\mathcal{R}, \mathcal{W})$ grows superlinearly in $|\mathcal{W}|$, preventing $\mathcal{M}$ from deciding membership in $\mathcal{L}(\mathcal{R})$ within $\mathcal{O}(|\mathcal{W}|)$ time (Su et al., 2024).

*Example* 3.1. Consider the regex: $\hat{} (.+?) \backslash 1 + \$$.

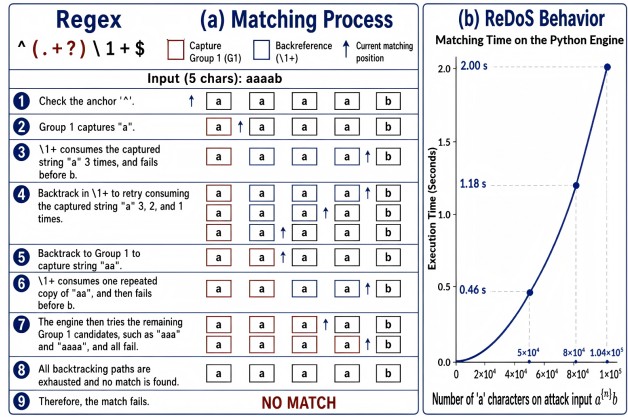

*Figure 1.* Matching process on the input $aaaab$ and ReDoS behavior on attack input $a^{\{n\}}b$ for the regex $\hat{}(.+?)\backslash 1+\$$.

This regex is extracted from PyPI (Wang et al., 2023). The subexpression $(.+?)$ is a capturing group that matches the shortest non-empty prefix of the input string. The subexpression $\backslash 1+$ (i.e., a backreference quantified by +) then requires the remaining suffix of the input to consist of one or more repetitions of exactly the same captured substring. The anchors $\hat{}$ and $\$$ constrain the match to the entire input. Taken together, the regex matches strings composed of repeated copies of an arbitrary non-empty substring.

Considering Example 3.1, the attack string $a^{\{n\}}b$ triggers ReDoS vulnerability. Since this string cannot be accepted, the regex engine performs a polynomial number of backtracking steps to enumerate all possible substrings in the matching process. As shown in Figure 1(a), on the input $aaaab$, the engine first captures $a$ by Group 1, lets $\backslash 1+$ match repeated copies of this captured string, and then checks the end anchor $\$$. The check fails because an unmatched suffix remains. The engine then backtracks by trying all possible strings for $\backslash 1+$ and, after no valid match is found, enlarges the captured substring, e.g., from $a$ to $aa$. In other words, the regex engine enumerates all candidate substrings for the capturing group, each of which is then tested against the backreference $\backslash 1+$. As shown in Figure 1(b), as the number of $a$ in string $a^{\{n\}}b$ increases, the matching time grows superlinearly on the Python engine.

**Heterogeneous Graph.** A heterogeneous graph (Sun & Han, 2012) is a directed graph consisting of multiple types of entities (nodes) and relations (edges), enabling the modeling of complex systems with diverse semantics. Formally, it is defined as follows.

**Definition 3.2** (Heterogeneous Graph). A heterogeneous graph is a tuple $\mathcal{G} = (\mathcal{V}, \mathcal{E}, \mathcal{A}, \mathcal{R})$, where $\mathcal{V}$ and $\mathcal{E}$ denote the sets of nodes and edges, respectively, and $\mathcal{A}$ and $\mathcal{R}$ represent the sets of node types and edge types. Each node $v \in \mathcal{V}$ and edge $e \in \mathcal{E}$ are associated with type mapping functions $\tau(v) : \mathcal{V} \to \mathcal{A}$ and $\phi(e) : \mathcal{E} \to \mathcal{R}$, respectively. A

graph is considered heterogeneous if at least one of the node or edge sets contains more than one type, i.e., $|\mathcal{A}| + |\mathcal{R}| > 2$.

**Definition 3.3** (Abstract Syntax Tree (AST)). A regex is represented by an AST:

$$\mathcal{T} = (\mathcal{V}_T, \mathcal{E}_T) \tag{1}$$

where $\mathcal{V}_T$ is the set of nodes and $\mathcal{E}_T$ is the set of directed parent–child edges. Each node is labeled by a syntactic operator type via the labeling function $\text{op} : \mathcal{V}_T \to \mathcal{A}_T$. For any node $v \in \mathcal{V}_T$, we write $\text{op}(v)$ for its operator type. For any edge $(u, v) \in \mathcal{E}_T$, we say that $u$ is the parent of $v$ in the syntactic structure.

# 4. Method

In this section, we propose ReDoS-MotifGNN (RMGNN) (illustrated in Fig. 2) which takes the regex as input. Generally, the framework of the method consists of three parts: **(i)** Construction of HRGs; **(ii)** Motif-Enhanced HRGs; **(iii)** Graph Representations Learning. In the following, we describe these parts in more detail.

## 4.1. Problem Formulation

We formalize the identification of ReDoS vulnerability as a binary classification problem: given a regex, the goal is to decide whether it is vulnerable to ReDoS. We define the dataset as $\{(r_i, y_i) \mid r_i \in \text{R}, y_i \in \mathcal{Y}\}, i \in \{1, 2, \ldots, n\}$, where R denotes a set of regexes and $\mathcal{Y} = \{0, 1\}^n$ denotes the label set with $1$ for ReDoS vulnerable and $0$ otherwise. $n$ is the number of instances.

## 4.2. Construction of HRG

Since the AST preserves only the syntactic structure of a regex, we construct the HRG to model richer semantic node attributes and inter-node relations. Specifically, to better encode multidimensional node attributes under heterogeneity, we equip each node with an attribute vector and a type-specific binary mask, so that missing values are explicitly indicated rather than naively encoded as zeros. In addition, we augment the topology with semantic edges (e.g., *Capture-backreference*) to capture complex relations beyond the standard AST hierarchy (see Fig. 2(i)).

**Node-Level Semantic Augmentation.** Formally, let $\mathcal{T} = (\mathcal{V}_T, \mathcal{E}_T)$ be the AST of a regex (cf. Definition 3.3). We construct a HRG $\mathcal{G}_R = (\mathcal{V}_R, \mathcal{E}_R, \mathcal{A}_R, \mathcal{R}_R, \mathcal{X})$ by associating each AST node $v \in \mathcal{V}_T$ with a unique HRG node $u = \Psi(v) \in \mathcal{V}_R$ via a bijective mapping $\Psi : \mathcal{V}_T \to \mathcal{V}_R$. And $\mathcal{X}$ denotes the node attribute mapping, which extends the standard graph definition to explicitly encode the syntactic semantics of each node.

We define the node type mapping $\tau : \mathcal{V}_R \to \mathcal{A}_R$ and the attribute mapping $\mathcal{X} : \mathcal{V}_R \to \mathbb{R}^d$, where $d$ is the dimension of node attribute. These are induced from the AST structure and operator semantics via:

$$\tau(\Psi(v)) = \text{op}(v), \qquad \mathcal{X}(\Psi(v)) = f_{\text{attr}}(v), \tag{2}$$

where $f_{\text{attr}} : \mathcal{V}_T \to \mathbb{R}^d$ is a semantic analysis function. For each node $v$, it inspects the operator label $\text{op}(v)$ and the syntactic information associated with $v$, including counting bounds, capturing group indices, and lookaround types, to produce a comprehensive $d$-dimensional attribute vector.

Consequently, the attribute set is represented as a dense node feature matrix $\mathbf{X} \in \mathbb{R}^{|\mathcal{V}_R| \times d}$, constructed by stacking the processed vectors $\mathcal{X}(u)$ for all $u \in \mathcal{V}_R$. Detailed specifications regarding the feature space decomposition and specific encoding rules are provided in the Appendix C .

Finally, to represent such heterogeneous attributes in a unified graph-level format, we employ a masking mechanism. Let $t(u) \in \{0, 1\}^{|\mathcal{A}_R|}$ be the one-hot type vector of node $u$. We define a learnable type-attribute mask matrix $\mathcal{S} \in \{0, 1\}^{|\mathcal{A}_R| \times d}$. A node-specific mask is generated by $\mathcal{M}(u) = t(u)^\top \mathcal{S}$, and the effective attribute vector is obtained via element-wise masking:

$$\widetilde{\mathcal{X}}(u) = \mathcal{X}(u) \odot \mathcal{M}(u). \tag{3}$$

This mechanism ensures that if the $k$-th attribute dimension is irrelevant for type $\tau(u)$ (i.e., $\mathcal{M}_k(u) = 0$), it is ignored in subsequent processing, allowing the dense matrix $\mathbf{X}$ to efficiently represent sparse, type-specific semantics.

**Edge-Level Semantic Augmentation.** Beyond the parent–child edges of the AST, we introduce additional edges that encode semantic relations between nodes not fully captured by the AST structure. To represent syntactic and semantic information in a unified form, we model the HRG as a labeled multi-relational graph whose edge set is

$$\mathcal{E}_R \subseteq \mathcal{V}_R \times \mathcal{V}_R \times \mathcal{R}_R, \tag{4}$$

where each edge $(u, v, r) \in \mathcal{E}_R$ connects endpoints $u, v \in \mathcal{V}_R$ with relation type $r \in \mathcal{R}_R$.

Starting from the directed parent–child edges $\mathcal{E}_T$ in the AST, we construct $\mathcal{E}_R$ by assigning one or more relation labels to each AST edge via a rule-based generator $g$:

$$\{(u, v, r) \in \mathcal{E}_R \mid (u, v) \in \mathcal{E}_T, r \in g(u, v).\} \tag{5}$$

Here, $g(u, v) \subseteq \mathcal{R}_R$ returns the set of relation types induced by the operator/attribute context of $(u, v)$, as detailed below.

$$g : \mathcal{V}_T \times \mathcal{V}_T \to \wp(\mathcal{R}_R), \tag{6}$$

where $\wp$ denotes the power set. For each pair $(u, v)$, returns a (possibly empty) set $g(u, v) \subseteq \mathcal{R}_R$ determined by the operator types of $u$ and $v$. When $g(u, v) = \varnothing$, no additional edges are introduced between $u$ and $v$.

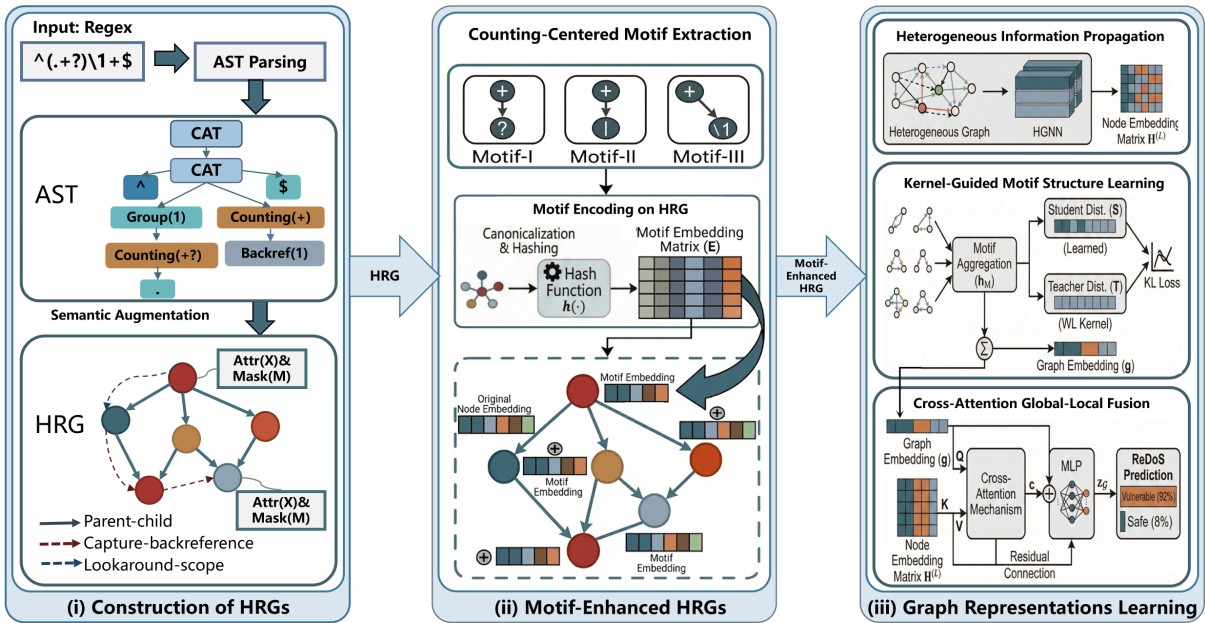

*Figure 2.* The overall framework of RMGNN, illustrated with Example 3.1. (i) We convert the regex AST into an HRG by augmenting it with multi-dimensional node attributes and additional edges. (ii) Counting-centered substructures (e.g., \1+) are extracted as motifs and encoded back into the HRG to enrich its structural semantics.(iii) The model utilizes heterogeneous propagation to update node embeddings, employs kernel-guided distillation to regularize motif structures, and fuses them via cross-attention for prediction.

The generator $g$ instantiates the additional edges according to the following semantic relations:

- *Lookaround-scope.* $g(u, v) = \texttt{LookArd}$, if the type of $u$ is lookaround and $v$ is a subexpression within the scope of $u$.

- *Capture-backreference.* $g(u, v) = \texttt{Ref}$, if the type of $u$ is capturing group and $v$ whose type is backreference refers to $u$.

These semantic relations enrich the HRG with non-parent-child dependencies, while preserving the syntactic backbone inherited from the AST.

### 4.3. Motif-Enhanced HRGs

In HRGs, certain substructures play a key role in graph classification, analogous to motifs in general graph domains (Wu et al., 2023; Yu & Gao, 2022). These substructures capture broadly representative structural regularities that correlate strongly with the backtracking behavior of regex engines. Specifically, neighborhoods centered around counting operators (i.e., $r^*$, $r^+$, $r^{\{m,\}}$, $r^{\{m,n\}}$, $r^{*?}$ and $r^{+?}$) are highly indicative of ReDoS. These constructs, widely used in regexes to increase regex expressiveness, can introduce ambiguity that can trigger excessive backtracking, thereby inflating runtime in regex engines (Cox, 2007; Wang et al., 2023; Siddiq et al., 2024b). To explicitly encode this ReDoS-related prior knowledge, we design three classes of

counting-centered motifs and propose a pipeline to extract and incorporate them into the HRG representation. Unlike the fine-grained handcrafted patterns used in prior work (Li et al., 2021; Parolini & Miné, 2023), our motifs are deliberately designed at a more abstract level: they do not encode specific vulnerability signatures but instead provide structurally indicative abstractions that allow the model to learn ReDoS-related motifs without being confined to overly local fragments. The overall process is visualized in Fig. 2(ii).

**Counting-centered Motifs Extraction.** Formally, let $\mathcal{G}_R$ be the base HRG. A motif instance $\mathcal{M}$ is an induced subgraph $\mathcal{G}_R[\mathcal{V}_\mathcal{M}]$ where $\mathcal{V}_\mathcal{M} \in \mathcal{V}_\mathcal{R}$. $\mathcal{G}_R[\mathcal{V}_\mathcal{M}]$ is centered at a counting node $v_{cnt}$ whose types are counting operators . We categorize these instances into three distinct classes:

*Motif-I: Nested and Sibling Counting Nodes.* In the HRG, this motif captures a substructure consisting of two counting nodes that are either *nested* or *siblings* in the hierarchical structure. Formally, a motif instance is identified for two counting nodes $v_{cnt_1}$ and $v_{cnt_2}$ if (i) $v_{cnt_2}$ lies in the descendant set of $v_{cnt_1}$ (or vice versa), i.e., there exists a directed path from one to the other; or (ii) they share the same parent node and thus occur at the same level in the HRG.

*Motif-II: Counting Node with Union Successor.* In the HRG, this motif captures a substructure where a counting node has a *Union*-type (e.g., $r|r$, [C]) node among its immediate successors. For example, the pattern $(a|aa)^*$ allows the matcher to choose between two branches that can consume

the same input prefix (e.g., the leading char $a$), leading to excessive matching paths.

*Motif-III: Extended Features under Counting Nodes.* In the HRG, this motif captures a substructure where a descendant set of a counting node has (i) nodes with lookaround types and/or (ii) nodes with backreference type. Such extended features introduce non-regular matching semantics and can significantly amplify matching cost under counting. For example, the subexpression \1+ in Example 3.1 exhibits this motif, as it contains a backreference under counting.

We construct motif-enhanced HRG by extracting these counting-centered motifs and embed them back into the HRG (see details in Appendix D).

### 4.4. Graph Representations Learning

To enable effective representation learning on HRGs, we propose ReDoS-MotifGNN (RMGNN). The framework is designed to capture both local and global structural information in HRGs for detecting ReDoS vulnerabilities. First, we employ heterogeneous information propagation to capture fine-grained syntactic dependencies. Second, observing that standard global pooling often fails to preserve the discriminative structural features of ReDoS vulnerabilities, we introduce a motif-level pooling mechanism regularized by graph kernel. This mechanism explicitly grounds the graph representation in the structural priors defined in Section 4.3. Finally, to synthesize a comprehensive representation that balances global motif-level semantics with fine-grained node-level representations, we implement a cross-attention fusion module. We detail each component below.

**Heterogeneous Information Propagation.** To learn structural priors across the motif-enhanced HRG, we employ heterogeneous information propagation, utilizing the motif-enhanced features $\mathbf{h}'_v$ and $\mathbf{e}'_{uv}$ (Appendix D.2) as the initial states $\mathbf{h}_v^{(0)}$ and $\mathbf{e}_{uv}^{(0)}$.

Based on the above node representations, we perform $L$ layers of propagation. At each layer $l$, the representation of node $v_i$ is updated by aggregating messages from its neighbors $\mathcal{N}_r(v_i)$ under specific relations $r \in \mathcal{R}_R$:

$$\mathbf{h}_i^{(l+1)} = \sigma\left(\sum_{r \in \mathcal{R}_R} \sum_{v_j \in \mathcal{N}_r(v_i)} \mathbf{W}_r^{(l)} \psi_r\left(\mathbf{h}_j^{(l)}, \mathbf{h}_i^{(l)}, \mathbf{e}_{(j,i,r)}^{(l)}\right)\right),$$

(7)

where $\sigma(\cdot)$ is an activation function, and $\mathcal{N}_r(v_i)$ denotes the neighbor set of $v_i$ under relation $r$. $\mathbf{W}_r^{(l)}$ is a relation-specific weight matrix, while the message function $\psi_r(\cdot)$ fuses the edge feature $\mathbf{e}_{(j,i,r)}^{(l)}$ with the node pair $(\mathbf{h}_j^{(l)}, \mathbf{h}_i^{(l)})$.

This process yields the final node embeddings $\mathbf{H}^{(L)} = \{\mathbf{h}_i^{(L)}\}_{v_i \in \mathcal{V}_R}$ (visualized in Fig. 2(iii)), which encode both fine-grained regex semantics and motif-level information.

**Kernel-Guided Motif Structure Learning.** Building on node representations learned from HRGs, we first aggregate them into motif-level embeddings via:

$$\mathbf{h}_{\mathcal{M}} = \text{READOUT}_{\text{motif}}\left(\{\mathbf{h}_v^{(L)} \mid v \in \mathcal{V}_{\mathcal{M}}\}\right). \quad (8)$$

To regularize these embeddings with structural priors, we employ the Weisfeiler-Lehman (WL) Subtree Kernel (Shervashidze et al., 2011) as a "teacher". The WL kernel $\mathcal{K}_{\text{WL}}$ iteratively aggregates discrete neighborhood labels to measure the fine-grained local structural similarity within $k$ hops. We align the pairwise similarity distribution of the learned continuous embeddings, denoted as the Student Distribution ($\mathbf{S}$), with the discrete structural similarity defined by the kernel, denoted as the Teacher Distribution ($\mathbf{T}$).

$$\begin{aligned} \mathbf{S}_{ij} &= \frac{\exp\left(\mathbf{h}_{\mathcal{M}_i}^\top \mathbf{h}_{\mathcal{M}_j}/\tau\right)}{\sum_k \exp\left(\mathbf{h}_{\mathcal{M}_i}^\top \mathbf{h}_{\mathcal{M}_k}/\tau\right)}, \\ \mathbf{T}_{ij} &= \frac{\exp\left(\mathcal{K}_{\text{WL}}(\mathcal{M}_i, \mathcal{M}_j)/\tau\right)}{\sum_k \exp\left(\mathcal{K}_{\text{WL}}(\mathcal{M}_i, \mathcal{M}_k)/\tau\right)}. \end{aligned} \quad (9)$$

where $\tau$ is a temperature parameter that controls the sharpness of the distribution.

Structural prior transfer is achieved by minimizing the KL-divergence (Kullback & Leibler, 1951) between these row-normalized distributions:

$$\mathcal{L}_{\text{distill}} = \sum_i \text{KL}(\mathbf{T}_{i\cdot}||\mathbf{S}_{i\cdot}) = \sum_{i,j} \mathbf{T}_{ij} \log \frac{\mathbf{T}_{ij}}{\mathbf{S}_{ij}}. \quad (10)$$

The motif embeddings, regularized by this structural constraint, are then aggregated to form the global motif representation $\mathbf{g} = \sum_{\mathcal{M} \in \mathcal{D}} \mathbf{h}_{\mathcal{M}}$.

**Cross-Attention Global-Local Fusion.** Naive concatenation of global and local features often results in redundancy. To address this, we employ a Cross-Attention mechanism to dynamically fuse these representations. Specifically, the global motif vector $\mathbf{g}$ acts as the *Query*, seeking relevant details from the node embeddings $\mathbf{H}^{(L)}$ which serve as *Keys* and *Values*. This allows the model to selectively focus on the fine-grained syntactic tokens that support the global structural judgment, effectively filtering out noise.

The attention weights are computed as:

$$\boldsymbol{\alpha} = \text{softmax}\left(\frac{(\mathbf{g}\mathbf{W}_Q)(\mathbf{H}^{(L)}\mathbf{W}_K)^\top}{\sqrt{d_k}}\right), \quad (11)$$

where $\mathbf{W}_Q, \mathbf{W}_K \in \mathbb{R}^{d \times d_k}$ are learnable projection matrices and $d_k$ is the dimension of the key. Unlike standard pooling, this mechanism generates an attended node representation $\mathbf{c} = \sum_{i \in \mathcal{V}} \alpha_i(\mathbf{h}_i^{(L)}\mathbf{W}_V)$. The final representation is fused via a residual connection to preserve gradient flow:

$$\mathbf{z}_{\mathcal{G}} = \text{MLP}(\mathbf{g} \parallel \mathbf{c}) + \mathbf{g}. \quad (12)$$

The fused graph representation $\mathbf{z}_{\mathcal{G}}$ is fed into an MLP classifier followed by a sigmoid function to produce the prediction. The model is trained end-to-end using binary cross-entropy.

# 5. Experiments

In this section, we conduct a series of experiments to evaluate the ReDoS detection capability of RMGNN. We aim to answer six questions: **Q1.** Does RMGNN achieve the best performance in ReDoS detection on real-world datasets when compared to six state-of-the-art baselines? **Q2.** How do different model baselines and RMGNN under different graph representations perform in ReDoS detection? **Q3.** How do the two inner components of RMGNN contribute to its effectiveness? **Q4.** How does RMGNN perform in efficiency evaluations? **Q5.** How does RMGNN perform when deployed to detect ReDoS vulnerabilities reported publicly through CVEs? **Q6.** How does RMGNN compare with LLM baselines in terms of detection performance? Additionally, we provide further analyzes on hyperparameter sensitivity and motif expressiveness in Appendix E.3 and Appendix E.5, respectively.

## 5.1. Experiment setup

**Datasets.** We evaluate RMGNN on four real-world datasets. The first is the corpus introduced by (Chapman & Stolee, 2016), which contains 13,597 regexes extracted from public Python repositories. The other three datasets, PyPI[1], Maven[2], and NuGet[3], are constructed by (Wang et al., 2023) and contain 118,099, 136,643, and 49,366 regexes, respectively, mined from the corresponding package repositories.

**Engines.** As the four datasets originate from three different programming language repositories (i.e., Corpus and PyPI from Python, Maven from Java, and NuGet from .NET), we conduct our experiments on Corpus and PyPI using the Python 3.8.10 regex library, on Maven using the Java 1.8.0_432 regex library, and on NuGet using the .NET 7.0 regex library.

**Baselines.** To comprehensively evaluate the performance of RMGNN, we identified six state-of-the-art ReDoS detection tools from prior works, categorized into: **(i)** static analysis (RegexStatic (Weideman et al., 2016) and Rexploiter (Wüstholz et al., 2017)); **(ii)** dynamic analysis (ReScue (Shen et al., 2018) and Regulator (McLaughlin et al., 2022)); **(iii)** hybrid approach (ReDoSHunter (Li et al., 2021) and Rengar (Wang et al., 2023)).

**Ground Truth.** To obtain reliable ground truth, we first run all six baselines on the four datasets and collect the union of their reported candidate vulnerable regexes. For each candidate, we run scripts to validate the baseline-generated payloads on the corresponding regex engine, following the criterion proposed by Su et al. (Su et al., 2024). Then, for the remaining regexes for which none of the baselines can

---

[1] https://pypi.org/

[2] https://maven.apache.org/

[3] https://www.nuget.org/

produce an effective payload, we further perform manual validation to establish the ground truth. Specifically, three authors independently inspect and validate these regexes on the target engine, and any inconsistent judgments are discussed until a consensus is reached. Inspired by the work (Wang et al., 2023), to reduce the manual effort, we use scripts to extract local subexpressions (LSes) from each regex and prioritize validating these LSes, which often suffice to determine whether the regex is ReDoS-prone.

**Implementation Details.** We implement RMGNN using the PyTorch library and conduct all experiments on a workstation equipped with an Intel Core i9-14900K CPU and an NVIDIA RTX 4090 GPU. Since ReDoS detection is highly engine-dependent, training and evaluation are performed under the supervision of the regex labels which encode the linear or superlinear backtracking behaviors exhibited by each engine in practice. Due to space constraints, detailed hyperparameter settings and training configurations are provided in the Appendix E.1. In our evaluation, we employ three key metrics: Accuracy (Acc), Recall (Rec), and the F1-score (F1). Let $TP$, $TN$, $FP$, and $FN$ denote the number of True Positives, True Negatives, False Positives, and False Negatives, respectively. **Acc** measures the overall proportion of correctly classified instances, calculated as $\frac{TP+TN}{TP+TN+FP+FN}$. Rec assesses the model's ability to identify relevant instances and is defined as $\frac{TP}{TP+FN}$. Finally, to provide a balanced view of performance by accounting for both false positives and false negatives, we utilize the F1, formulated as $\frac{2TP}{2TP+FP+FN}$. We set a per-regex timeout of 6 minutes for RMGNN inference and all baselines.

## 5.2. Main Results and Analysis

To answer **Q1**, we compare RMGNN against six state-of-the-art baselines, with results summarized in Table 1. As shown, RMGNN consistently achieves the best Acc and F1 across all four real-world datasets. In particular, compared with the second-best baseline on each dataset (mostly Regulator), RMGNN improves the F1 by 4.49%, 22.97%, 27.39%, and 28.06% on Corpus, PyPI, Maven, and NuGet, respectively, corresponding to relative gains of 6.13%, 52.1%, 73.99%, and 88.18%. Overall, RMGNN outperforms baselines in 14 out of 16 evaluation cases across all datasets and metrics, highlighting its comprehensive effectiveness.

Moreover, especially on the large-scale Maven dataset with over 100k regexes, RMGNN achieves over 70% relative improvement in F1 over the second-best baseline, together with substantial gains in TP and recall, demonstrating strong generalization at scale. Notably, on Corpus, ReDoSHunter trades a marginal 1.9% Recall gain for severe drops in Accuracy ($-50.2\%$) and F1 ($-43.0\%$), likely due to its high computational overhead when handling complex structures. In contrast, RMGNN maintains comparable recall while

*Table 1.* Comparison results between RMGNN and the baselines on the four real-world datasets. The best results are highlighted in **bold**, and the second best are underlined.

| Method | Corpus (Python) | | | | PyPI (Python) | | | | Maven (Java) | | | | NuGet (.NET) | | | |
|---|---|---|---|---|---|---|---|---|---|---|---|---|---|---|---|---|
| | TP | Acc | Rec | F1 | TP | Acc | Rec | F1 | TP | Acc | Rec | F1 | TP | Acc | Rec | F1 |
| RegexStatic | 12 | 69.46 | 0.29 | 0.57 | 78 | 77.72 | 0.30 | 0.59 | 108 | 80.45 | 0.42 | 0.81 | 62 | 82.95 | 0.75 | 1.45 |
| Rexploiter | 139 | 70.57 | 3.38 | 6.50 | 574 | 78.09 | 2.19 | 4.25 | 411 | 80.88 | 1.58 | 3.10 | 230 | 83.52 | 2.77 | 5.35 |
| ReScue | 16 | 69.80 | 0.39 | 0.77 | 88 | 77.89 | 0.34 | 0.67 | 821 | 81.18 | 3.16 | 6.09 | 139 | 83.41 | 1.68 | 3.28 |
| Regulator | 2,754 | 85.20 | 66.89 | 73.24 | 8,106 | 82.56 | 30.97 | 44.05 | 7,302 | 81.54 | 28.08 | 37.02 | 2,485 | 78.42 | 29.98 | 31.82 |
| ReDoSHunter | **3,056** | 43.50 | **74.23** | 44.31 | **16,937** | 23.95 | 64.72 | 27.39 | 10,298 | 30.71 | 39.60 | 18.09 | 4,171 | 30.66 | 50.31 | 19.59 |
| Rengar | 1,194 | 70.60 | 29.00 | 37.40 | 4,300 | 64.23 | 16.43 | 16.91 | 3,127 | 66.71 | 12.02 | 12.25 | 1,916 | 73.66 | 23.11 | 22.76 |
| **RMGNN (Ours)** | 2,999 | **87.36** | 72.84 | **77.73** | 17,069 | **85.78** | 65.22 | **67.02** | 19,949 | **83.62** | 76.71 | **64.41** | 5,823 | **84.19** | 70.24 | 59.88 |

*Table 2.* Comparison results between the RMGNN with other models and RMGNN variants (varying representations and components). The best results are highlighted in **bold**. RMGNN (MEHRG) denotes the full model using MEHRG (motif-enhanced HRG).

| Method | Corpus (Python) | | | | PyPI (Python) | | | | Maven (Java) | | | | NuGet (.NET) | | | |
|---|---|---|---|---|---|---|---|---|---|---|---|---|---|---|---|---|
| | TP | Acc | Rec | F1 | TP | Acc | Rec | F1 | TP | Acc | Rec | F1 | TP | Acc | Rec | F1 |
| Bi-LSTM | 1,363 | 72.12 | 33.11 | 41.84 | 6,158 | 68.50 | 23.53 | 24.87 | 6,649 | 63.85 | 25.57 | 21.47 | 2,264 | 72.31 | 27.31 | 24.88 |
| Tree-LSTM | 1,772 | 76.96 | 43.04 | 53.09 | 6,510 | 78.08 | 24.88 | 33.47 | 7,108 | 64.54 | 27.33 | 22.95 | 2,458 | 73.10 | 29.65 | 27.02 |
| Transformer | 1,738 | 75.28 | 42.22 | 50.85 | 7,742 | 72.98 | 29.58 | 32.67 | 7,566 | 65.22 | 29.09 | 24.43 | 2,782 | 71.87 | 33.56 | 28.61 |
| SAT | 1,534 | 68.75 | 37.26 | 41.94 | 8,974 | 75.06 | 34.29 | 37.87 | 6,879 | 64.20 | 26.45 | 22.21 | 2,588 | 71.09 | 31.22 | 26.61 |
| GCN (AST) | 1,704 | 78.31 | 41.39 | 53.62 | 7,918 | 70.28 | 30.26 | 31.09 | 8,484 | 66.58 | 32.62 | 27.39 | 3,882 | 81.41 | 46.83 | 45.83 |
| GCN (HRG) | 1,909 | 75.45 | 46.37 | 53.35 | 9,678 | 70.26 | 36.98 | 35.53 | 10,318 | 69.31 | 39.68 | 33.31 | 4,076 | 80.67 | 49.17 | 46.07 |
| GCN (MEHRG) | 2,181 | 81.21 | 52.98 | 63.07 | 11,438 | 73.25 | 43.71 | 42.00 | 10,547 | 69.65 | 40.56 | 34.05 | 4,205 | 77.64 | 50.72 | 43.24 |
| GAT (AST) | 1,261 | 67.68 | 30.63 | 36.47 | 8,094 | 70.58 | 30.93 | 31.78 | 8,254 | 66.24 | 31.74 | 26.65 | 3,105 | 73.18 | 37.45 | 31.93 |
| GAT (HRG) | 1,431 | 71.65 | 34.76 | 42.61 | 9,502 | 71.46 | 36.31 | 36.06 | 9,630 | 68.28 | 37.03 | 31.09 | 3,558 | 75.02 | 42.92 | 36.59 |
| GAT (MEHRG) | 1,602 | 78.28 | 38.91 | 52.05 | 10,734 | 73.55 | 41.02 | 40.73 | 10,089 | 68.97 | 38.80 | 32.57 | 3,946 | 76.59 | 47.60 | 40.58 |
| Graphormer (AST) | 1,670 | 74.28 | 40.56 | 48.86 | 8,182 | 72.22 | 31.26 | 33.28 | 8,575 | 66.72 | 32.97 | 27.68 | 3,817 | 76.06 | 46.04 | 39.25 |
| Graphormer (HRG) | 1,789 | 76.03 | 43.45 | 52.34 | 10,030 | 72.06 | 38.33 | 37.81 | 10,433 | 69.48 | 40.12 | 33.68 | 4,140 | 77.37 | 49.94 | 42.57 |
| Graphormer (MEHRG) | 1,874 | 76.40 | 45.52 | 53.88 | 11,262 | 72.95 | 43.03 | 41.35 | 12,152 | 72.03 | 46.73 | 39.23 | 4,464 | 78.69 | 53.85 | 45.90 |
| GraphGPS (AST) | 1,840 | 76.20 | 44.69 | 53.21 | 8,534 | 72.52 | 32.61 | 34.47 | 8,828 | 67.09 | 33.95 | 28.50 | 3,836 | 76.14 | 46.27 | 39.44 |
| GraphGPS (HRG) | 2,147 | 80.71 | 52.15 | 62.09 | 10,505 | 75.26 | 40.14 | 41.83 | 11,006 | 70.33 | 42.32 | 35.53 | 4,354 | 78.24 | 52.52 | 44.77 |
| GraphGPS (MEHRG) | 2,284 | 82.73 | 55.48 | 66.05 | 12,493 | 81.02 | 47.74 | 52.72 | 13,872 | 74.59 | 53.34 | 44.79 | 4,658 | 79.47 | 56.19 | 47.90 |
| RMGNN (AST) | 2,045 | 80.39 | 49.67 | 60.54 | 8,270 | 76.87 | 31.60 | 37.71 | 8,942 | 67.26 | 34.39 | 28.87 | 3,687 | 79.09 | 44.48 | 41.67 |
| RMGNN (HRG) | 2,249 | 82.21 | 54.63 | 65.04 | 13,070 | 73.39 | 49.94 | 45.41 | 10,777 | 69.99 | 41.44 | 34.79 | 4,529 | 78.95 | 54.63 | 46.57 |
| **RMGNN (MEHRG)** | **2,999** | **87.36** | **72.84** | **77.73** | **17,069** | **85.78** | **65.22** | **67.02** | **19,949** | **83.62** | **76.71** | **64.41** | **5,823** | **84.19** | **70.24** | **59.88** |
| w/o kernel | 2,386 | 82.46 | 57.95 | 66.69 | 13,549 | 82.81 | 51.77 | 57.17 | 15,133 | 76.46 | 58.19 | 48.86 | 5,176 | 81.57 | 62.44 | 53.22 |
| w/o attention | 2,590 | 84.29 | 62.91 | 70.80 | 14,957 | 79.20 | 57.15 | 54.92 | 15,363 | 76.80 | 59.08 | 49.60 | 5,240 | 81.83 | 63.21 | 53.88 |

delivering markedly better overall performance. Static baselines (RegexStatic and Rexploiter) exhibit low number of TPs and recall, largely because they have limited capability in handling extended features in real-world regexes.

To answer **Q2**, we evaluate RMGNN under three graph representations and compare it with a broad set of learning-based baselines on the four benchmarks, including sequence/tree-based models (Bi-LSTM and Tree-LSTM), transformer-based baselines (Transformer and SAT), GNNs (GCN and GAT), and modern graph transformers (Graphormer and GraphGPS), as reported in Table 2. Under the MEHRG representation, RMGNN consistently achieves the best F1 scores, outperforming the best-performing baseline by 11.68%, 14.30%, 19.62%, and 11.98% on Corpus, PyPI, Maven, and NuGet, respectively. Moreover, under the same HRG representation, RMGNN still outperforms a standard GCN in terms of F1-score with an average F1 gain of 5.9% across the four datasets. This

suggests that the proposed GNN design remains effective even without motif enhancement. Finally, replacing AST with MEHRG yields a 25.1% average F1 gain, demonstrating the benefit of enriched semantics and motif annotations.

To answer **Q3**, we evaluate two ablations in Table 2: (1) *w/o kernel*, which removes the kernel alignment, and (2) *w/o attention*, which replaces the cross-attention fusion with naive mean pooling. *w/o kernel* leads to an average 10.8% F1 drop driven by an average 13.7% Recall decline, highlighting its role in providing structural guidance to distinguish vulnerable motifs from similar non-vulnerable cases. This result indicates that the kernel-guided component is an important part of the RMGNN design. Similarly, *w/o attention* yields an average 10.0% F1 drop (up to 14.81% on large-scale datasets such as Maven), indicating that attention is crucial for capturing key motifs (e.g., backreferences) in complex ReDoS detection.

To answer **Q4**, we evaluate and compare the average time

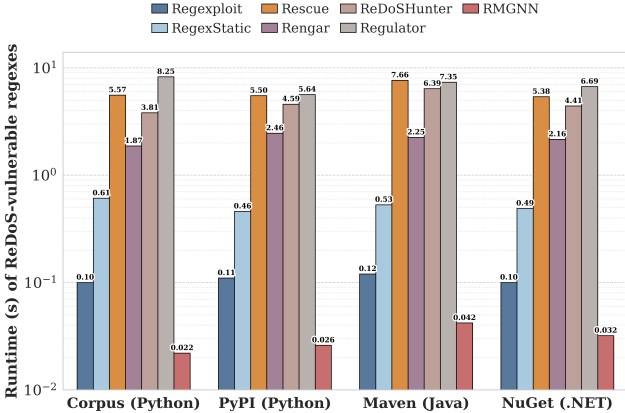

*Figure 3.* Efficiency Comparison on four datasets.

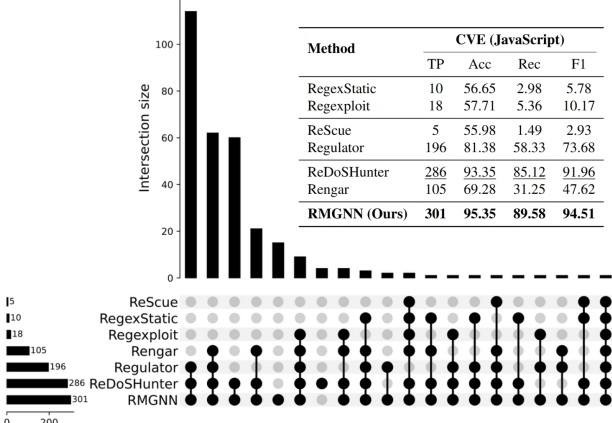

*Figure 4.* Intersection analysis and performance comparison on CVE (JavaScript).

required to detect vulnerable regexes of RMGNN and all baselines on the four benchmarks. As illustrated in Figure 3, while static tools maintain low latency, dynamic and hybrid approaches incur significantly higher overhead due to their dynamic execution on regex engines. In contrast, RMGNN achieves an average detection time of 0.031s via efficient GNN inference across the four datasets. This performance represents a speedup of at least 2.86× compared to the fastest-baseline Regexploiter. Furthermore, against the top F1-performing baseline Regulator, RMGNN achieves an average speedup of 244×, peaking at 375× on the Corpus dataset. This low latency makes RMGNN highly suitable for large-scale Continuous Integration pipelines.

To answer **Q5**, regarding exposed ReDoS vulnerabilities in the wild, we collect the reported CVEs from JavaScript repositories spanning 2015 to 2025. Based on these confirmed cases, we obtain a set of 340 vulnerable regexes, forming a realistic benchmark for evaluating zero-day detection. We then test these regexes with our trained model (RMGNN) as well as existing ReDoS baselines to assess

*Table 3.* Performance and efficiency comparison between RMGNN and LLM-based baselines on the **Corpus** dataset. Time denotes the inference time per sample.

| Method | TP | Acc | Rec | F1 | Time |
|---|---|---|---|---|---|
| GLM-4-Plus | 2,197 | 74.97 | 53.40 | 56.41 | 35.958s |
| QwQ-32B | 2,556 | 72.39 | 62.08 | 57.67 | 13.176s |
| QwQ-32B-AWQ | 2,830 | 68.33 | 68.74 | 56.80 | 9.142s |
| GPT-OSS-20B | 703 | 73.31 | 17.09 | 27.97 | 33.749s |
| **RMGNN (Ours)** | **2,999** | **87.36** | **72.84** | **77.73** | **0.022s** |

*Table 4.* Performance comparison between RMGNN and LLM-based baselines on the **PyPI** dataset.

| Method | TP | Acc | Pre | Rec | F1 |
|---|---|---|---|---|---|
| QwQ-32B | 13,763 | 53.74 | 24.58 | 52.59 | 33.50 |
| QwQ-32B-AWQ | 14,396 | 50.45 | 23.55 | 55.00 | 32.98 |
| Qwen3-Coder-Next | 15,088 | 56.08 | 27.00 | 57.65 | 36.78 |
| GPT-OSS-20B | 2,771 | 75.79 | 34.78 | 10.59 | 16.23 |
| **RMGNN (Ours)** | **17,069** | **85.78** | **68.92** | **65.22** | **67.02** |

whether they can identify real-world ReDoS vulnerabilities. As illustrated in Fig. 4, RMGNN outperforms all baselines, achieving 95.35% accuracy (F1: 94.51%) and identifying the largest number of ReDoS vulnerabilities, including all cases detected by the other five baselines. These results highlight its potential for discovering emerging ReDoS vulnerabilities in practical settings.

Finally, to answer **Q6**, we compare RMGNN with four representative LLM baselines on the Corpus and PyPI benchmarks, as shown in Tables 3 and 4. RMGNN consistently outperforms all LLM baselines on both datasets. In particular, on Corpus it achieves an F1-score of 77.73, exceeding the best-performing LLM baseline by over 20%, with only 0.022s inference time per sample, and on PyPI it improves Recall by 7.57–54.63 points, Accuracy by 9.99–35.33 points, Precision by 34.14–45.37 points, and F1 by 30.24–50.79 points over the LLM baselines. These results show that RMGNN is a far more practical solution for efficient ReDoS detection, and detailed LLM settings and further analysis are referred to Appendix E.2.

## 6. Conclusion

This paper proposes RMGNN, a motif-based graph learning framework for ReDoS detection. It constructs motif-enhanced HRGs to preserve extended features and encodes counting-centered motifs as structural priors. RMGNN utilizes heterogeneous propagation and kernel-guided motif learning, fused via cross-attention, to jointly model local semantics and global vulnerability patterns. Experiments on four real-world datasets demonstrate that RMGNN outperforms six state-of-the-art baselines in F1-score while delivering substantially faster inference speed.

## Acknowledgements

The authors would like to thank the editors and reviewers for their valuable comments. This work is supported by the National Natural Science Foundation of China (Grant No. 62372439) and the Natural Science Foundation of Beijing, China (Grant No. F251021).

## Impact Statement

This paper presents work whose goal is to advance the field of Graph Representation Learning for ReDoS Detection. There are many potential societal consequences of our work, none which we feel must be specifically highlighted here.

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

# A. Detailed Related Works

## A.1. ReDoS Detection

Previous works have studied ReDoS detection, which can be mainly classified as static analysis, dynamic analysis, hybrid approaches. Static analysis (Kirrage et al., 2013; Davis et al., 2018) inspects regex structures without execution, so it is efficient, but it often suffers from high false positives/negatives and provides limited support for extended features (e.g., lookarounds and backreferences). For instance, RXXR/RXXR2 (Kirrage et al., 2013) detects vulnerabilities via structural patterns, while RegexStatic (Weideman et al., 2016) and Rexploiter (Wüstholz et al., 2017) identify vulnerable NFA patterns with exponential or polynomial complexity; however, because these methods generally lack a mechanism to validate candidate attack strings, they are prone to false positives and may still fail to confirm ReDoS, as the generated strings are not guaranteed to be rejected by the target engine. Dynamic analysis (Petsios et al., 2017; Shen et al., 2018; McLaughlin et al., 2022) detects ReDoS vulnerabilities at run time and is generally considered more precise than static analysis. In practice, most dynamic tools rely on fuzzing: they repeatedly generate candidate inputs, execute them on a real regex engine to discover time-consuming strings, and then use the observed behavior to infer the regex's worst-case time complexity. Rescue (Shen et al., 2018) avoids purely random generation and instead searches for time-consuming inputs; however, given the enormous search space, it mainly identifies exponential or high-degree polynomial vulnerabilities, and may miss lower-degree polynomial cases or deeply hidden vulnerabilities. Regulator (McLaughlin et al., 2022) further develops a sophisticated mutation strategy to fuzz JavaScript regex bytecode. Despite their effectiveness, dynamic approaches often output a random attack string and provide limited insight into the root causes of the underlying ReDoS vulnerability. Finally, hybrid approaches aim to eliminate false positives by validating candidate strings on real regex engines, yet their effectiveness largely hinges on how well such candidate strings are generated. Revealer (Liu et al., 2021) generates candidates by leveraging Java's regex engine, while ReDoSHunter (Li et al., 2021) constructs candidates based on five predefined ReDoS patterns. Rengar (Wang et al., 2023) further attempts to produce rejected strings using hand-crafted rules to handle 'disturbances'. Nevertheless, these tools generally rely on predefined patterns or rules, thus limiting their effectiveness on a specific class of ReDoS vulnerability.

Besides, several learning-based studies leverage machine learning (ML) or large language models (LLMs) for related tasks. RegexClassifier (Lu et al., 2023) explores GNN-based regex analysis with NFA inputs, but its NFA-based formulation cannot directly handle non-regular extended features. Other studies include assessing LLM-generated regexes (Siddiq et al., 2024a), performing reactive ReDoS detection via resource usage monitoring (Demoulin et al., 2019), and generating PoCs (Simsek et al., 2025). However, these methods do not primarily target proactive, regex-level identification of ReDoS vulnerabilities. RMGNN is designed as a scalable detector for proactive vulnerability scanning. It learns regex structures through a novel learning model that supports extended features, while concrete attack exploitation can be delegated to downstream dynamic/hybrid tools when needed.

## A.2. GNNs for Vulnerability Detection

Graph Neural Networks (GNNs) have attracted significant attention due to their strong capability to capture both structural and semantic information in programs. Early work, such as Code Property Graphs (CPG) (Yamaguchi et al., 2014), integrates Abstract Syntax Trees (AST), Control Flow Graphs (CFG), and Data Flow Graphs (DFG) into a unified representation, enabling joint reasoning over program structure. In terms of graph encoders, GGNN (Allamanis et al., 2018) introduced gated message passing for program graphs, while models such as R-GCN (Schlichtkrull et al., 2018) and HGT (Hu et al., 2020) proposed principled mechanisms to handle multi-relational and heterogeneous graph data. Building on these foundations, Devign (Zhou et al., 2019) leveraged composite graph representations to achieve accurate vulnerability detection. Subsequent works, including HELoC (Wang et al., 2022) and heterogeneous program graph approaches (Zhang et al., 2022), further demonstrated that explicitly modeling hierarchical structures and typed information can significantly enhance code representation learning and semantic understanding. To improve both effectiveness and interpretability, later studies explored deeper and bidirectional GNN architectures (e.g., DeepWukong (Cheng et al., 2021), BGNN4VD (Cao et al., 2021)), dataflow-inspired learning (Steenhoek et al., 2024), statement-level detection (Hin et al., 2022), and explanation techniques including counterfactual reasoning (Wen et al., 2023; Chu et al., 2024; Cao et al., 2024).

More recently, the transition from traditional message-passing GNNs to Graph Transformers has enabled better modeling of long-range dependencies. In particular, Heterogeneous Graph Transformers (HGTs) have been successfully applied to software vulnerability detection (e.g., DSHGT (Zhang et al., 2024a)) and Just-in-Time vulnerability detection (e.g., HgtJIT (Sun et al., 2025)). Furthermore, models like MANDOO-HGT (Nguyen et al., 2023) and GTrans (Zhang et al.,

*Table 5.* Summary of Rule-Based Attribute Encoding for HRG Nodes. The "Active Dimensions" correspond to the non-zero entries in the type-dependent mask $\mathcal{M}(u)$. Node types are aligned with the semantic analysis defined in Section 4.2.

| Node Type $\tau(u)$ | Attribute Dimensions | Description & Semantic Alignment |
|---|---|---|
| Counting Operator | Boundedness Flag 
 Bounds $(m, n)$ 
 Greediness Flag | Indicates if the operator represents bounded counting (1 if $m, n < \infty$). 
 Normalized values of lower/upper bounds (preserving $r\{m, n\}$ structure). 
 1 for greedy matching (priority), 0 for lazy. |
| Group | Capture Flag 
 Group Index | 1 for capturing groups, 0 for non-capturing clusters. 
 Ordinal index $k$ (corresponds to *capturing group indexs*). |
| BackReference | Target Index | The index of the capturing group being referenced. |
| Lookaround | Lookaround Type 
 Polarity | Categorical: Lookahead vs. Lookbehind. 
 1 for positive assertion, 0 for negative logic. |
| Character Class | Class Category 
 Cardinality | One-hot encoding: `Digit`, `Word`, `Space`, or `Custom`. 
 Approximate size of the character set (e.g., $|[a-z]| = 26$). |

2024b) extend these capabilities to specific domains, targeting smart contract vulnerability detection and obfuscation-resilient binary code similarity, respectively. Regular expressions, similar to executable programs, exhibit hierarchical AST structures, control flow, and rich semantic relations. Inspired by graph-based program analysis and advances in heterogeneous graph learning, we model regex ASTs as heterogeneous graphs and employ relation-aware GNNs to detect ReDoS vulnerabilities. This demonstrates that graph-based reasoning can be effectively extended to structured pattern languages.

## B. Regular Expressions

Let $\Sigma$ be a finite alphabet, a string is a finite sequence of characters over $\Sigma$, $\Sigma^*$ is the set of all strings over $\Sigma$, $\varepsilon$ is the empty word, $\varnothing$ is the empty set, $\mathbb{N}$ is the set of natural numbers, and $\infty$ is infinity. Classical regular expressions over $\Sigma$ are defined as

$$r ::= \quad \varepsilon \mid \varnothing \mid a \mid [\mathrm{C}] \mid r|r \mid rr \mid r^{\{m,n\}} \mid r^? \mid r^+ \mid r^*,$$

where $a \in \Sigma$, $\mathrm{C} \subseteq \Sigma$, and $m \le n$, $n \in \mathbb{N} \cup \{\infty\}$. Common shorthands include $r^? = r^{\{0,1\}}$, $r^* = r^{\{0,\infty\}}$, $r^+ = r^{\{1,\infty\}}$, $r^{\{m\}} = r^{\{m,m\}}$, and $r^{\{m,\}} = r^{\{m,\infty\}}$.

Modern regular expression engines extend classical syntax with non-regular constructs, enabling more powerful and flexible pattern matching. The formal grammar for modern regular expressions with extended features is as follows:

$$\begin{aligned} r ::= \quad & (r)_i \mid \backslash i \mid (?=r) \mid (?!r) \mid (?<=r) \mid (?<!r) \\ & \mid r^{??} \mid r^{*?} \mid r^{+?} \mid \hat{} \mid \$ \mid \backslash b \mid \backslash \mathrm{B}, \end{aligned}$$

where $(r)_i$ denotes the $i$-th capturing group, $\backslash i$ is a backreference to that group, $(?=r)$ and $(?!r)$ are positive and negative lookaheads, $(?<=r)$ and $(?<!r)$ are positive and negative lookbehinds, $\hat{}$ and $\$$ are start and end anchors, and $\backslash b$ / $\backslash \mathrm{B}$ denote word and non-word boundaries, respectively. The lazy quantifiers $r^{??}$, $r^{*?}$, and $r^{+?}$ match as few characters as necessary while still satisfying the overall pattern.

## C. Rule-Based Semantic Analysis for HRG Attributes

Complementing the formal definition in Section 4.2, the heterogeneous node attributes are generated via the semantic analysis function $f_{\mathrm{attr}}$. This function maps each node $u$ to a dense feature vector $\mathcal{X}(u)$ based on its operator type $\tau(u)$ and syntactic context. As described in Eq. (2), a learnable mask matrix $\mathcal{S}$ is employed to handle attribute heterogeneity, ensuring that only dimensions relevant to $\tau(u)$ are activated. Table 5 summarizes the mapping between node types and their corresponding active attribute dimensions.

**Bounded Counting Attributes.** As highlighted in Section 4.2, preserving **bounded counting operators** without expansion is crucial for optimizing graph scale. For a counting operator node $v$ (e.g., $r\{m, n\}$), $f_{\mathrm{attr}}(v)$ extracts:

- **Bounds** $(m, n)$: The specific lower bound $m$ and upper bound $n$ are encoded directly. This avoids the structural redundancy of expanding repetitions into concatenated sub-trees in the AST. Infinite bounds are clamped to a constant $C_{\max}$.

---

**Algorithm 1** Construction of Heterogeneous Regex Graph (HRG)

---

1: **Input:** Regex $r$, Semantic function $f_{\text{attr}}$, Mask matrix $\mathcal{S}$
2: **Output:** $\mathcal{G}_R = (\mathcal{V}_R, \mathcal{E}_R, \mathcal{A}_R, \mathcal{R}_R, \mathcal{X})$
3: $\mathcal{T} = (\mathcal{V}_T, \mathcal{E}_T) \leftarrow \text{ParseAST}(r);$ Init $\mathcal{V}_R, \mathcal{E}_R, \mathcal{A}_R, \mathcal{X} \leftarrow \emptyset$
4: **for each** node $v \in \mathcal{V}_T$ **do**
5: $\quad u \leftarrow \Psi(v);\ \tau(u) \leftarrow \text{op}(v);\ \mathcal{V}_R \leftarrow \mathcal{V}_R \cup \{u\};\ \mathcal{A}_R \leftarrow \mathcal{A}_R \cup \{\tau(u)\}$
6: $\quad \mathcal{X}(u) \leftarrow f_{\text{attr}}(v) \odot (\text{OneHot}(\tau(u))^\top \mathcal{S})$
7: **end for**
8: **for each** edge $(v_i, v_j) \in \mathcal{E}_T$ **do**
9: $\quad \mathcal{E}_R \leftarrow \mathcal{E}_R \cup \{(\Psi(v_i), \Psi(v_j), \texttt{Child})\}$
10: **end for**
11: **for each** pair $(u, v) \in \mathcal{V}_R \times \mathcal{V}_R$ **do**
12: $\quad R_{sem} \leftarrow \emptyset$
13: $\quad$ **if** Lookaround $(u) \wedge v \in \text{Scope}(u)$ **then** $R_{sem} \leftarrow R_{sem} \cup \{\texttt{LookArd}\}$
14: $\quad$ **if** Group $(u) \wedge v$ is $\text{Ref}(u)$ **then** $R_{sem} \leftarrow R_{sem} \cup \{\texttt{Ref}\}$
15: $\quad$ **for** $r \in R_{sem}$ **do**
16: $\quad\quad \mathcal{E}_R \leftarrow \mathcal{E}_R \cup \{(u, v, r)\}$
17: $\quad$ **end for**
18: **end for**
19: **return** $\mathcal{G}_R$

---

- **Greediness**: A binary flag indicating the matching priority, differentiating greedy operators from lazy ones.

**Group and Structural Attributes.** To support the generation of semantic edges such as *Capture-backreference* and *Lookaround-scope*, the attributes encode specific syntactic indices:

- **Group Index**: For capturing group nodes, the explicit ordinal index is recorded. This value enables the model to align the group definition with downstream BackReference nodes.

- **Lookaround Semantics**: Nodes are distinguished by their Lookaround Type and Polarity. These attributes provide the necessary context for the *Lookaround-scope* edges defined in the edge generator $g(u, v)$.

**Terminal Semantics.** For leaf nodes (Character Classes), semantics are encoded via Class Category and Cardinality. The cardinality attribute approximates the branching factor of the underlying state machine, enriching the semantic representation of the basic building blocks.

# D. Motif Extraction and Encoding

## D.1. Motif Extraction

Given an HRG $\mathcal{G}_R = (\mathcal{V}_R, \mathcal{E}_R)$, let $\mathcal{V}_{cnt} \subseteq \mathcal{V}_R$ denote the set of counting nodes. For each $v \in \mathcal{V}_{cnt}$, we identify all connected induced subgraphs within its local neighborhood that match the counting-centered motifs defined in Section 4.3.

To derive a permutation-invariant representation, we apply a canonicalization function $\text{Canon}(\cdot)$. Specifically, $\text{Canon}(\cdot)$ performs a deterministic breadth-first search (BFS) starting from the motif center(e.g., a counting node). To construct the sequence, neighboring nodes are ordered lexicographically by their node types (and edge relations) to ensure uniqueness. We record the sequence of structural attributes visited, yielding a canonical sequence $S_\mathcal{M} = \text{Canon}(\mathcal{M})$. To efficiently index these motifs without maintaining a growing dictionary, we employ feature hashing to map each sequence to a fixed-size vocabulary $\mathcal{V}_{\text{mot}} = \{1, \ldots, K\}$. The motif index is computed as:

$$ID_\mathcal{M} = \text{Hash}(S_\mathcal{M}) \bmod K. \tag{13}$$

For each HRG, we represent the extracted motifs as a set $\mathcal{D} = \{(c_m, \mathcal{V}_m)\}$, where each tuple pairs a canonical index $c_m$ with the node set $\mathcal{V}_m$ inducing the instance.

---

**Algorithm 2** Counting-Centered Motif Extraction and Encoding

---
1: **Input:** HRG $\mathcal{G}_R$, Embeddings **E**;     **Output:** $\mathbf{H}'$
2: $\mathbf{Z} \leftarrow \mathbf{0};$     $\mathcal{V}_{cnt} \leftarrow \{v \in \mathcal{V}_R \mid \text{Type}(v) \in \text{CountingOps}\}$
3: **for all** $c \in \mathcal{V}_{cnt}$ **do**
4:     $\mathcal{M}_1 \leftarrow \{u \in \mathcal{V}_{cnt} \mid u \in \text{Desc}(c) \cup \text{Sib}(c)\}$
5:     $\mathcal{M}_2 \leftarrow \{u \in \text{Child}(c) \mid \text{Type}(u) = \text{UNION}\}$
6:     $\mathcal{M}_3 \leftarrow \{u \in \text{Desc}(c) \mid \text{Type}(u) \in \{\text{LOOK}, \text{REF}\}\}$
7:     $\mathcal{S}_c \leftarrow \{\text{InducedSubgraph}(\{c, u\}) \mid u \in \mathcal{M}_1 \cup \mathcal{M}_2 \cup \mathcal{M}_3\}$
8:     **for all** $\mathcal{M} \in \mathcal{S}_c$ **do**
9:         $k \leftarrow \text{Hash}(\text{Canon}(\mathcal{M})) \bmod K$
10:         $\mathbf{Z}[v] \leftarrow \mathbf{Z}[v] + \mathbf{E}[k]$ **for each** $v \in \mathcal{V}(\mathcal{M})$
11:     **end for**
12: **end for**
13: $\mathbf{h}'_v \leftarrow \mathbf{h}_v \parallel \phi(\mathbf{Z}[v])$ **for all** $v \in \mathcal{V}_R$
14: **return** $\mathbf{H}'$

---

## D.2. Motif Encoding on HRG

We initialize a learnable embedding matrix $\mathbf{E} \in \mathbb{R}^{K \times d_m}$ to represent the global motif vocabulary, where $K$ denotes motif vocabulary size and $d_m$ denotes the motif embedding dimension. For each instance $\mathcal{M}$, we retrieve its embedding as a vector $\mathbf{E}[ID_{\mathcal{M}}]$. These embeddings provide a compact representation of motif-level structural features.

To incorporate this structural prior into the HRG, we augment the node-level and edge-level representations with motif embeddings. Specifically, for each node $v$ (or edge $e$), we accumulate the embeddings of all motif instances that contain it. The updated representations are defined as:

$$\mathbf{h}'_v = \mathbf{h}_v \parallel \phi_v \left( \sum_{\mathcal{M} \in \mathcal{D}:\, v \in \mathcal{V}_{\mathcal{M}}} \mathbf{E}[ID_{\mathcal{M}}] \right), \tag{14}$$

$$\mathbf{e}'_{uv} = \mathbf{e}_{uv} \parallel \phi_e \left( \sum_{\mathcal{M} \in \mathcal{D}:\, (u,v) \in \mathcal{E}_{\mathcal{M}}} \mathbf{E}[ID_{\mathcal{M}}] \right), \tag{15}$$

where $\mathbf{h}_v$ and $\mathbf{e}_{uv}$ denote the initial features for node $v$ and edge $(u, v)$, respectively. $\mathcal{D}$ denotes the set of all extracted motif instances in the graph. The operator $\parallel$ represents concatenation, and $\phi_v(\cdot), \phi_e(\cdot)$ are linear projections mapping motif embeddings to the target feature space.

This design integrates high-order structural features into local representations, effectively complementing the fine-grained semantic attributes encoded in the original HRG.

# E. Detailed experiments

## E.1. Implementation Details

We implement RMGNN using the PyTorch and PyTorch Geometric libraries. Regarding the model architecture, we employ a three-layer heterogeneous information propagation structure with a hidden dimension size of 128. The model is trained using the Adam optimizer with an initial learning rate of $1e^{-3}$ and a weight decay of $5e^{-4}$ for regularization. We set the batch size to 64 and train for a maximum of 100 epochs, employing an early stopping mechanism based on validation loss to prevent overfitting. All experiments are conducted on ubuntu20.04 equipped with an Intel 14900k 24 Cores 32 Threads CPU and an NVIDIA RTX 4090 GPU. Since ReDoS detection is highly engine-dependent, our implementation follows the corresponding native regex engines of the datasets during labeling and evaluation, as described in the engines and ground truth paragraphs of Section 5.1. In this way, the supervision signals inherently encode the linear or superlinear backtracking behaviors exhibited by each engine in practice. Consequently, RMGNN is trained to predict regex-level vulnerabilities under engine-validated supervision, rather than from engine-agnostic annotations alone.

### E.2. LLM Evaluation Details

**Baselines and Dataset Selection.** To evaluate the performance of RMGNN against Generative AI, we selected a diverse set of Large Language Models (LLMs) representing different architectures and optimization strategies. Specifically, we include: (1) **GLM-4-Plus**, Zhipu AI's proprietary model optimized for complex reasoning and coding via large-scale alignment; (2) **QwQ-32B** is the reasoning model of the Qwen series, designed for improved thinking and reasoning on hard problems, while (3) **QwQ-32B-AWQ** is **QwQ-32B**'s AWQ-quantized 4-bit variant for efficient deployment with a full 131,072-token context window; and (4) **GPT-OSS-20B**, a medium-sized open-weight model for low latency, local, or specialized use cases (21B parameters with 3.6B active parameters). Given the significant computational overhead of LLMs, we adopted a high-fidelity evaluation strategy by selecting the Corpus dataset.

**LLM Deployment and Inference.** For the closed-source LLM baseline (GLM-4-Plus), we use the official API provided by the vendor. For the open-source LLM baselines (e.g., QwQ-32B, QwQ-32B-AWQ, and GPT-OSS-20B), we deploy them locally on a single machine running Ubuntu 24.04.2 LTS (noble) equipped with $4\times$ NVIDIA L40 GPUs (46 GB VRAM per GPU). The system uses NVIDIA driver 560.35.03 and CUDA 12.6, as verified by nvidia-smi. The software environment is managed via mamba and includes Python 3.10.19, vLLM 0.13.0, PyTorch 2.9.0, Transformers 4.57.3, and ModelScope 1.33.0. Model weights and configurations are downloaded in advance through ModelScope and stored locally under ./runLLM/models, avoiding any runtime dependence on external repositories.

All open-source LLM baselines are served using vLLM's OpenAI-compatible HTTP server with 4-way tensor parallelism and a maximum of 32 concurrent sequences. We issue non-streaming asynchronous requests to the /v1/chat/completions endpoint and enforce JSON-formatted outputs. Additionally, we adopted the official default generation hyperparameters (e.g., temperature) provided in their model configurations. These setups reflect out-of-the-box performance and ensure a fair, reproducible evaluation across all baselines.

**Experimental Results Analysis.** Table 3 and Figure 5 reveals a significant performance gap between RMGNN and generative models. RMGNN achieves an F1-score of 77.73, surpassing the best LLM (QwQ-32B) by over 20 points. This disparity reflects a common limitation in LLM-based vulnerability detection: hallucinations. Unlike RMGNN's strict learning-based classifier, LLMs often rely on superficial text patterns, leading to erroneous judgments. This is evident in the quantized QwQ-32B-AWQ, where a trade-off (Accuracy -4.06%, Recall +6.66% compared to the QwQ-32B) suggests that quantization noise amplifies uncertainty, causing the model to aggressively "hallucinate" threats in benign code (false positives). Conversely, the domain-specific GPT-OSS-20B fails to understand complex backtracking behaviors (Recall 17.09), indicating a lack of deep logical reasoning. Finally, with a speedup of $415\times$ to $1600\times$ over the LLMs, RMGNN proves to be a far more practical solution for efficient ReDoS detection.

To further examine whether this advantage generalizes to larger and more complex benchmarks, we additionally extend the comparison to the **PyPI** dataset, which contains 118,099 regexes. Since full-scale evaluation on larger datasets is financially and computationally expensive, especially for commercial APIs and locally deployed LLMs, our initial LLM study focused on Corpus. Here, we further include **Qwen3-Coder-Next**, a recently released open-weight coding model built on the Qwen3-Next architecture with hybrid attention and sparse MoE. As shown in Table 4, RMGNN remains clearly superior on this larger benchmark, improving Recall by 7.57–54.63 points, Accuracy by 9.99–35.33 points, Precision by 34.14–45.37 points, and F1 by 30.24–50.79 points over the LLM baselines. These results indicate that LLM-based methods still suffer from substantial false positives and unstable structural reasoning, whereas RMGNN is markedly more reliable for ReDoS detection.

### E.3. Hyperparameter Sensitivity Analysis

In this section, we perform sensitivity analyses to evaluate the impact of key hyperparameters on the proposed RMGNN architecture, using two representative datasets: Maven and Corpus.

**Impact of Propagation Layers.** Figure 6 presents the performance of RMGNN with different numbers of propagation layers, ranging from 1 to 5. RMGNN achieves the best performance when $L$ is set to 3. Fewer layers ($L = 1, 2$) yield relatively lower F1-scores, indicating that shallow propagation fails to capture the long-range dependencies between distant nodes in the HRG. Specifically, increasing $L$ from 1 to 3 results in a significant performance boost, raising the F1-score by

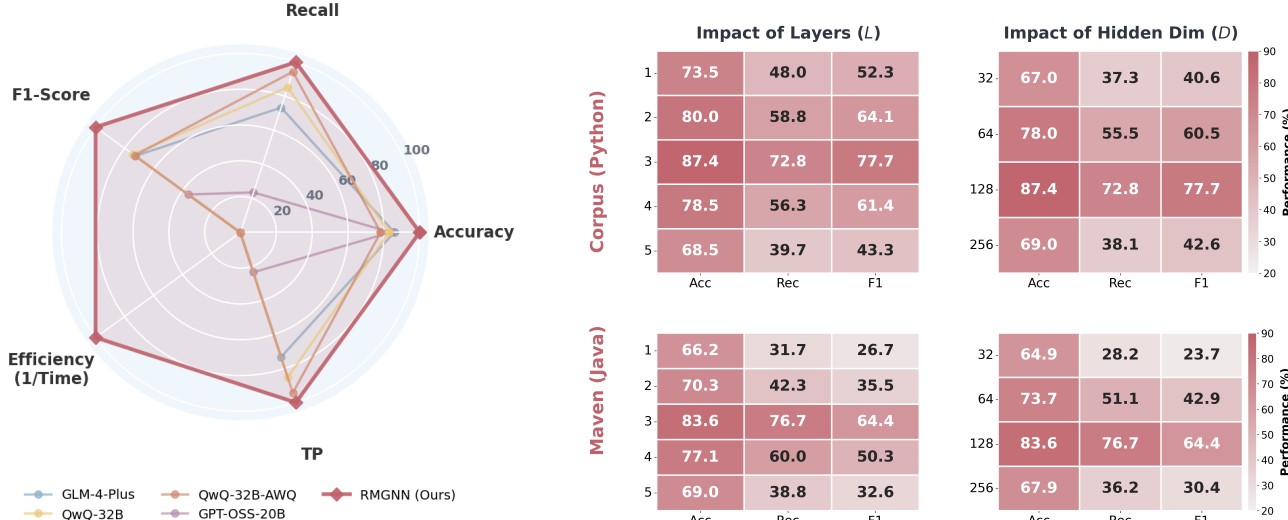

*Figure 5.* Performance comparison on the Corpus dataset. All metrics are normalized to a relative scale of [0, 100]. Efficiency is calculated as the reciprocal of inference time (1/Time).

*Figure 6.* Hyperparameter sensitivity analysis of RMGNN on Corpus and Maven datasets. The figure visualizes the impact of layers ($L$) and hidden dimension ($D$).

25.40 points on Corpus and 37.76 points on Maven. This confirms that a sufficient receptive field is essential for learning global structural patterns. However, performance degrades as $L$ increases further. When $L$ reaches 5, the F1-score on Corpus drops sharply to 43.30. We attribute this decline to the over-smoothing issue, where excessive aggregation causes node representations to become indistinguishable, thereby reducing the model's discriminative capability. To alleviate this limitation, we incorporate a residual connection in the final fusion module to preserve gradient flow and improve training stability. Although it does not fully eliminate performance degradation under all hyperparameter settings, it helps stabilize the practical configuration used in Fig. 6.

**Impact of Hidden Dimension Size.** We also investigate the effect of the hidden dimension size on model performance. As shown in the bottom section of Figure 6, RMGNN benefits from a larger parameter space to encode semantic features. The model achieves optimal results with $D = 128$. Increasing $D$ from 32 to 128 brings substantial gains, improving the F1-score by over 37 points on both datasets. This suggests that a sufficient dimension size is necessary to encompass the diverse attributes of heterogeneous regex nodes. Conversely, an excessively large dimension ($D = 256$) leads to a notable performance drop (e.g., F1 decreases to 42.63 on Corpus). This degradation indicates that an overly complex parameter space induces overfitting particularly given the structural constraints of the regex data.

To address the inherent over-smoothing issue common in GNN architectures, we introduce a residual connection in the final fusion module, which preserves gradient flow and improves training stability. Although this does not unconditionally resolve degradation across all theoretical hyperparameter spaces, it practically stabilizes the configurations evaluated in our sensitivity analysis (Figure 6). Achieving robust generalization under wider hyperparameter ranges warrants further investigation in this domain.

### E.4. Zero-Day Generalization without Motif Enhancement

To further examine whether zero-day generalization relies on explicit motif enhancement, we additionally compare RMGNN (HRG) with the full model, RMGNN (MEHRG), on 340 real-world zero-day ReDoS cases collected from JavaScript CVEs. As shown in Table 6, even without motif enhancement, RMGNN (HRG) already achieves strong zero-day performance, indicating that the model still retains substantial generalization ability under the HRG representation alone. Compared with the full model, the performance drop is moderate: TP decreases by 18, accuracy by 2.40 points, recall by 5.35 points, and F1 by 3.07 points. These results suggest that zero-day generalization does not rely solely on motif enhancement, while also

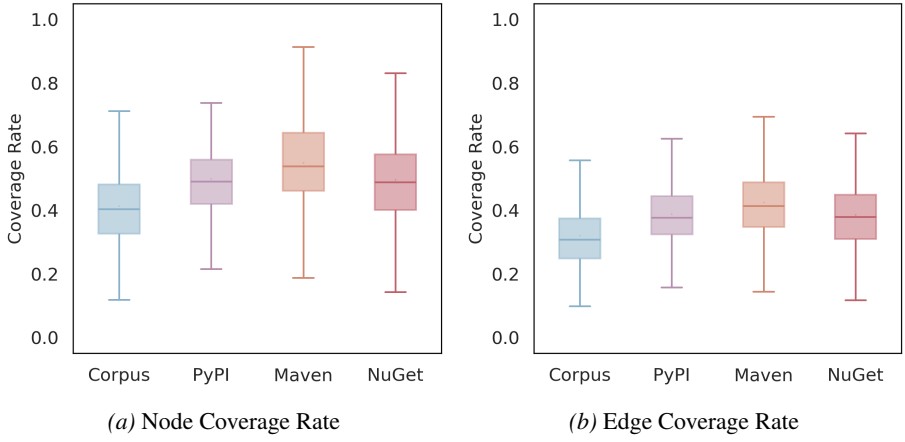

*(a)* Node Coverage Rate        *(b)* Edge Coverage Rate

*Figure 7.* The distributions of NCR and ECR across four datasets.

*Table 6.* Comparison of RMGNN variants on zero-day ReDoS cases from JavaScript CVEs.

| Method | TP | Acc | Rec | F1 |
|---|---|---|---|---|
| RMGNN (HRG) | 283 | 92.95 | 84.23 | 91.44 |
| RMGNN (MEHRG) | 301 | 95.35 | 89.58 | 94.51 |

confirming that motif enhancement provides a consistent additional gain.

### E.5. Statistical Analysis of Motif Structural Expressiveness

A core design principle of RMGNN is the explicit encoding of "counting-centered motifs" (Section 4.3) as structural priors. To demonstrate that these motifs represent fundamental topological components of real-world regexes rather than ad-hoc rules for specific vulnerabilities, we conduct a statistical coverage analysis across all four datasets. Inspired by subgraph coverage pattern mining, we define two key metrics to quantify motif prevalence: **Node Coverage Rate (NCR)**, which measures the proportion of nodes in an HRG belonging to at least one identified motif instance, and **Edge Coverage Rate (ECR)**, which measures the proportion of edges involved in the internal structure of these motifs. High values in these metrics indicate that the proposed motifs successfully encapsulate a significant portion of the regex's control flow into high-level semantic units, thereby validating their role as a representative structural abstraction.

Figure 7 shows the NCR and ECR distributions across the Corpus, PyPI, Maven, and NuGet datasets. The median NCR consistently ranges between 35% and 55%, demonstrating that our three motif classes successfully capture the core ReDoS-related structural patterns. This broad coverage acts as a strong structural prior, enabling RMGNN to generalize effectively, especially on large datasets like Maven, by focusing on these key risk factors rather than noise. Furthermore, the ECR (25%–45%) indicates that our method effectively summarizes local details, allowing the model to better capture long-range interactions. This approach overcomes the limitations of static tools, which suffer from limited recall due to strict pattern definitions. The consistency of these coverage rates across diverse datasets suggests that our motifs represent universal structural features, ensuring robust performance for cross-engine ReDoS detection.

Figure 8 visualizes the t-SNE embeddings of different models to interpret their underlying feature learning capabilities. The baseline methods, especially GCN variants using basic ASTs, exhibit severe entanglement where vulnerable (red) and benign (blue) instances overlap significantly. This visual data explains the performance disparity noted in Table 2, as the lack of clear boundaries suggests these models rely on class distribution biases rather than learning robust structural features. Consequently, their high accuracy scores are misleading and fail to translate into effective F1 performance.

In contrast, RMGNN utilizing the MEHRG representation demonstrates superior discriminative capability by forming compact clusters with distinct decision margins. The model successfully disentangles the complex regex manifold and separates vulnerable patterns from benign ones. This clear structural distinction confirms that RMGNN effectively captures high-level semantic features, such as vulnerable motifs, which are essential for ReDoS detection. These visual results strongly validate the significant F1-score improvements and the generalization ability reported in our quantitative experiments.

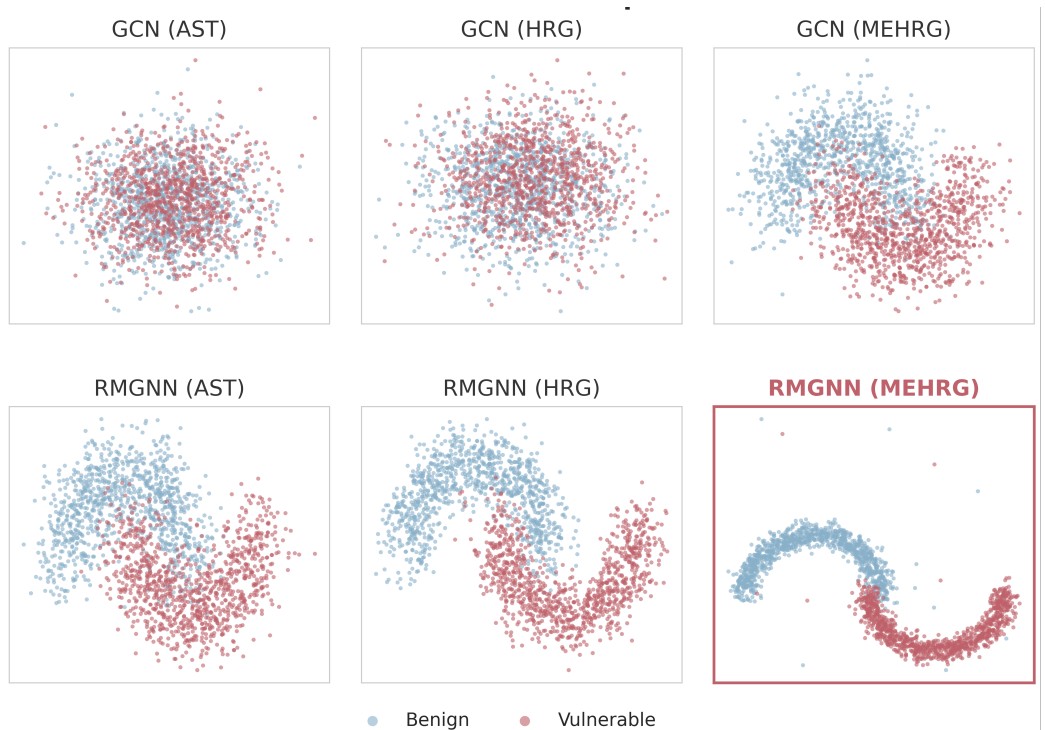

*Figure 8.* t-SNE visualization of learned embeddings for the ablation study on the Corpus dataset.

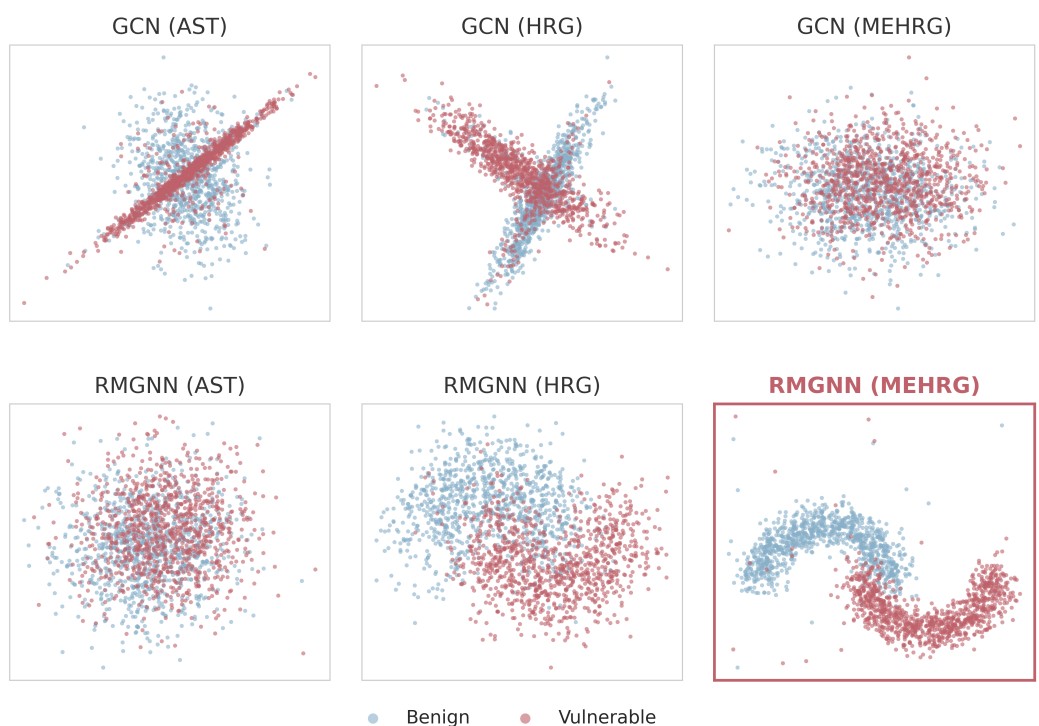

*Figure 9.* t-SNE visualization on the more challenging PyPI dataset. The separation is less pronounced reflecting lower F1-scores, yet RMGNN still outperforms baselines.

