# OpenReview forum: "Exploring Motif-based Heterogeneous Graph Learning for ReDoS Detection"
_ICML.cc/2026/Conference — ICML 2026 regular_

### Official Review · Reviewer_ANcd · 2026-03-07

**Soundness:** 3
**Presentation:** 3
**Significance:** 3
**Originality:** 3
**Overall Recommendation:** 5
**Confidence:** 3

**Summary:**

This paper addresses the problem of proactively detecting Regular Expression Denial-of-Service (ReDoS) vulnerabilities at the regex level, arguing that existing approaches (static analysis, fuzzing etc.) are not tailored for this task. The authors propose a graph neural network framework that: converts regex Abstract Syntax Trees (ASTs) into motif-enhanced Heterogeneous Regex Graphs (HRGs); extracts counting-centered motifs and encodes them as structural priors; and applies heterogeneous propagation with kernel-guided distillation as a structural regularizer, and cross-attention fusion for final classification. Experiments are conducted on four real-world regex datasets and an additional CVE JavaScript dataset, showing that RMGNN outperforms six baselines in F1-score while being substantially faster at inference.

**Compliance With Llm Reviewing Policy:**

Affirmed.

**Final Justification:**

While I remain concerned that the kernel may introduce too strong a structural prior, I appreciate the authors’ additional experiments.

**Key Questions For Authors:**

1. The ablation "without kernel" in Table 2 shows large drops in F1 score across all four datasets, which could raise a question on the stability of the model - is it highly sensitive to this kernel-guided distillation regularizer?

2. If the computation of the pairwise WL kernel features is crucial to the improvement, what fraction of training time does this "teaching" mechanism actually consumes?

3. All baselines from Table 1 are not learning-based models, which means the paper is comparing a learned model against a set of algorithmic ones. Table 2 partially examines whether the RMGNN architecture is the appropriate design among alternatives, but the compared models (Bi-LSTM, GCN variants, etc.) are relatively basic. Even within the GNN context, how would a Graph Attention Network perform on the same HRG?

4. Meanwhile, as LLM evaluation is conducted on the Corpus dataset (Appendix E.2), it is the smallest dataset and also the dataset where the gap between RMGNN and traditional non-learning-based baselines is already the narrowest (Table 1). It's the scenario where LLMs struggle most relative to inductive methods. Could the authors extend the LLM comparison to larger dataset (e.g. PyPI) to confirm the advantage holds when more training data is available?

**Limitations:**

Yes. There could be a bit more discussions on the binary classification without severity quantification, generalizability to syntax or semantics not seen from training, interpretability in specific vulnerability patterns detected by the model, etc.

**Strengths And Weaknesses:**

Strength:

- Framing ReDoS detection as a graph classification problem over motif-enhanced HRGs is a natural but compelling idea. The observation of counting-centered substructures serving as vulnerability indicators is well-grounded.

- The HRG construction is thoughtful and goes meaningfully beyond a plain AST representation. Lookaround-scope and backreference edge types encode semantic information about regex structure.

- Experiments are conducted quite comprehensively, with ablation study isolating the effect of representational designs and model architectures.

- The code base is well-structured and consistent.

Weakness:

- No error bars or standard deviations are reported. Tables 1 and 2 report single numbers with no variance estimates from cross-validation - which should be straightforward from code output. These would be crucial for assessing whether the improvements over baselines (especially the closer ones) are statistically meaningful.

- The approach detects vulnerability but does not generate attack strings or quantify severity. Tools like ReDoSHunter and Rengar produce actual attack strings that demonstrate the vulnerability, which could be practically important.

- Inference-time efficiency has been compared on four datasets, but the computational cost of the WL kernel during training has not been discussed.

---

> ### Author Rebuttal · Authors · 2026-03-30
>
> We sincerely thank reviewer ANcd for the careful reading and constructive feedback.
>
> **W1: Variance reporting**
>
> **RW1:** Variance estimates are important for assessing statistical significance. In the revision, we will report 10-fold cross-validation results (mean ± std.) for all learning-based models in Tables 1 and 2.
>
> **W2 \& Limitations: Severity quantification and attack-string generation**
>
> **RW2:** Our method makes a trade-off by prioritizing scalability over attack string generation related to dynamic validation. This design favors scalability and efficiency: RMGNN serves as a scalable detector for large-scale and real-time scenarios, while concrete attack construction can be delegated to downstream dynamic or hybrid tools when needed. We will make this limitation explicit in the revision and expand the discussion on generalizability and interpretability, including a case study in the appendix.
>
> **Q1 : Stability of the Model**
>
> **RQ1:** The w/o kernel result is better understood from the perspective of model design. In RMGNN, Kernel-Guided Motif Structure Learning regularizes motif embeddings with local structural similarity and explicitly grounds motif-level representations in HRG structural priors. Without this mechanism, the model separate structurally similar but behaviorally different cases only through learned continuous representations, which is substantially harder. The consistent gain across all four datasets therefore highlights the effectiveness of the proposed kernel-guided design and its importance in the full RMGNN formulation. As you suggest, we will clarify this point more explicitly in the revision.
>
> **W3 \& Q2: Computational cost of the WL kernel during training**
>
> **RW3 \& RQ2:** The training-time cost of the WL kernel will be clarified more explicitly. In our framework, Kernel-Guided Motif Structure Learning serves as the teaching mechanism for the distillation loss, within which the pairwise WL-kernel computation is used to provide structural supervision. As shown in Table 1, the full teaching mechanism (including WL-kernel computation) accounts for only 8.77\%--10.86\% of the total training time across the four datasets, indicating that it introduces only a small one-time cost during training and does not dominate the overall optimization process. Since this component is used only during training and discarded at inference time, it does not affect deployment efficiency. Overall, the WL kernel serves as a training-time structural regularizer rather than a prohibitive computational bottleneck.
>
> ### Table 1.
> |Dataset|Teaching mechanism(incl.WL-kernel)(s)|Total training time(s)|Teaching mechanism/Total|
> |-|-|-|-
> |Corpus|59.69|581.28|10.27\%
> |PyPI|589.31|5425.95|10.86\%
> |Maven|608.06|5702.01|10.66\%
> |NuGet|282.37|3218.22|8.77\%
>
> **Q3: Comparison to Stronger Graph Learning Baselines**
>
> **RQ3:** Detailed quantitative comparisons with recent learning-based models are provided in our response to Q1 \& Q2 \& W3 raised by Reviewer pJ9L (Stronger Baselines). To further address the reviewer’s question within the GNN context, we additionally evaluate GAT under the same three input representations. The results show that GAT is a relatively strong graph-learning baseline, especially under MEHRG, but it still remains clearly below RMGNN.
>
> ### Table 2.
> |Method|Corpus||||PyPI||||
> |-|-|-|-|-|-|-|-|-
> ||TP|Acc|Rec|F1|TP|Acc|Rec|F1
> |GAT(AST)|1,261|67.68|30.63|36.47|8,094|70.58|30.93|31.78
> |GAT(HRG)|1,431|71.65|34.76|42.61|9,502|71.46|36.31|36.06
> |GAT(MEHRG)|1,602|78.28|38.91|52.05|10,734|73.55|41.02|40.73
>
> **Q4: Extending the LLM comparison on PyPI**
>
> **RQ4:** Our initial LLM evaluation focused on Corpus because full-scale evaluation on larger datasets is financially and computationally expensive, especially for commercial APIs and locally deployed LLMs. To address this point, we additionally extend the comparison to the larger PyPI benchmark (118,099 regexes) and will add the results to Appendix E.2. We also include Qwen3-Coder-Next, a recently released open-weight coding model built on the Qwen3-Next architecture with hybrid attention and sparse MoE. The updated results in Table 3 show that RMGNN remains clearly superior on this larger and more complex dataset: it improves Recall by 7.57-54.63 points, Accuracy by 9.99-35.33 points, Precision by 34.14-45.37 points, and F1 by 30.24-50.79 points. These results indicate that LLM-based methods still suffer from substantial false positives and unstable structural reasoning, whereas RMGNN is markedly more reliable for ReDoS detection.
>
> ### Table 3.
> |Method|TP|Acc|Pre|Rec|F1|
> |-|-|-|-|-|-
> |QwQ-32B|13,763|53.74|24.58|52.59|33.50
> |QwQ-32B-AWQ|14,396|50.45|23.55|55.00|32.98
> |Qwen3-Coder-Next|15,088|56.08|27.00|57.65|36.78
> |GPT-OSS-20B|2,771|75.79|34.78|10.59|16.23
> |**RMGNN**|**17,069**|**85.78**|**68.92**|**65.22**|**67.02**
>
> If your concerns have been addressed, could you kindly consider raising your score? We greatly appreciate your comments and support.

---

> > ### Author Rebuttal · Reviewer_ANcd · 2026-04-03
> >
> > From another perspective, the w/o kernel results may suggest that the kernel introduces a strong structural prior that promotes robust interpolation within the observed motif space, while potentially limiting extrapolation to truly novel, out-of-distribution structures. Such priors also tend to yield more conservative predictions, often reducing false positives; while this can improve robustness, it may also lead to less expressiveness.
> >
> > That said, I appreciate the additional experiments provided by the authors, and I will raise my score accordingly. Some of the supplementary results could remain in the appendix, but I would recommend moving Table 3 here into the main text.

---

> > > ### Author Response · Authors · 2026-04-03
> > >
> > > We sincerely thank you for your careful reading of our rebuttal. We are pleased that the rebuttal helped clarify your concerns and led to a consensus.
> > >
> > > Based on your valuable suggestions, we will include Table 3 in the main text of the revised manuscript and place the other supplementary results in the appendix. In addition, some of your comments highlight meaningful directions that deserve further investigation in future work.

---

### Official Review · Reviewer_XxiN · 2026-03-10

**Soundness:** 3
**Presentation:** 2
**Significance:** 2
**Originality:** 2
**Overall Recommendation:** 4
**Confidence:** 2

**Summary:**

The authors propose a GNN-based Regex Denial-of-Service attack detection method, called RMGNN.

Specifically, RMGNN models given Regex as heterogeneous graphs, and apply graph neural network to obtain vector representations of this graph.

Experiments demonstrate the effectiveness of the proposed method over existing baseline methods.

**Compliance With Llm Reviewing Policy:**

Affirmed.

**Final Justification:**

The authors have addressed my concerns. However, given that I have limited knowledge in this field, I would like to maintain my score: weak-acceptance. But still, myself leans toward the acceptance.

**Key Questions For Authors:**

Refer to weakness section.

**Limitations:**

Refer to weakness section.

**Strengths And Weaknesses:**

### Strength

- S1) Proposed RMGNN shows strong performance in benchmark datasets.
- S2) Proposed RMGNN shows faster inference speed, which is crucial for the real-world applications.

### Weakness

- W1) It seems that the proposed method covers three motifs. Then, can all the motifs within the heterogeneous graph can be expressed with the three motifs? If there are certain motifs we cannot take into account, what is the (practical) issue?

- W2) For people who have limited understanding in this domain may feel difficulty in understanding the concept of ReDos. Therefore, I recommend the authors to include some visual descriptions for the high-level concept of ReDos.

---

> ### Author Rebuttal · Authors · 2026-03-30
>
> We sincerely thank reviewer XxiN for detailed reading and meaningful feedback.
>
> **QW1: Motif Coverage and Model Robustness**
>
> **RW1:** To clarify, the three motifs (i.e., counting-centered substructures) are intended to cover all ReDoS-relevant structures considered in this paper. From the perspective of ReDoS detection, counting operators are the critical structural source of risk in real-world regexes, especially when combined with nested/sibling counting, union successors, or extended features such as lookarounds and backreferences. It is worth noting that the three counting-centered motifs are designed at an abstract level to cover ReDoS-relevant structures, rather than fine-grained patterns that can be directly used for ReDoS detection.
>
>
> **W2: Improving Readability with a Visual Explanation of ReDoS**
>
> **RW2:** We fully agree that the algorithmic complexity behind ReDoS may be difficult to understand for readers without prior background in ReDoS. Improving the accessibility of our paper to such readers is an important goal of the revision. To address this, we will add a high-level conceptual diagram to Section 3 (Preliminaries) of the revised manuscript. This visual description will intuitively illustrate the ReDoS mechanism by showing the execution flow of a regex engine together with a vulnerable structural pattern.
>
> Specifically, the diagram will use Example 3.1 from our paper, `^(.+?)\1+$`, together with a crafted malicious input such as $a^n b$ (e.g., $aaaa...ab$). The visual aid will highlight:
>
> - **The matching logic:** how the lazy capturing group `(.+?)` matches a candidate prefix, and how the subsequent backreference `\1+` attempts to match the remaining suffix.
> - **The ReDoS mechanism:** when the string cannot be accepted, the regex engine is forced to backtrack repeatedly and enumerate many candidate substrings, resulting in polynomial or exponential matching cost.
>
> By visually illustrating this catastrophic backtracking behavior, the added figure will make the high-level concept of ReDoS easier to understand for readers without prior ReDoS knowledge, and will also provide a clearer bridge from the intuitive attack mechanism to our later graph-based formulation.
>
> If your concerns have been addressed, could you kindly consider raising your score? We greatly appreciate your comments and support.

---

> > ### Author Rebuttal · Reviewer_XxiN · 2026-04-02
> >
> > Thank you for the responses.
> >
> > However, given that I have limited knowledge in this field, while I think the work is reasonable according to my academic guess, I feel hard to raise my score to strong acceptance.
> >
> > Thus, I would like to maintain my score: weak acceptance.

---

> > > ### Author Response · Authors · 2026-04-04
> > >
> > > We sincerely thank you for your careful consideration of our rebuttal. We fully understand and respect your decision, and greatly appreciate your positive assessment.

---

### Official Review · Reviewer_pJ9L · 2026-03-10

**Soundness:** 2
**Presentation:** 2
**Significance:** 2
**Originality:** 2
**Overall Recommendation:** 3
**Confidence:** 3

**Summary:**

This paper proposes RMGNN, a heterogeneous GNN for detecting ReDoS-vulnerable regular expressions. The approach builds a heterogeneous regex graph (HRG) from regex ASTs, explicitly modeling extended regex constructs (e.g., lookarounds, backreferences) and injecting counting-centered motif priors. The authors report strong results on four large datasets (317K regexes), with improved F1 and substantially faster inference than dynamic or hybrid analyzers.

**Compliance With Llm Reviewing Policy:**

Affirmed.

**Final Justification:**

I have decided to raise my score from 2 to 3 as a response to the authors' rebuttal.

**Key Questions For Authors:**

1. More recent technical comparisons are needed, and the experimental environment should be up-to-date with the latest technology.

2. Detailed data is needed to support the advantages compared to LLM or static methods.

3. A more clearly defined innovative contribution is needed, rather than simply applying GNN to a traditional threat scenario.

**Limitations:**

yes

**Strengths And Weaknesses:**

Strengths:
* Scale and practicality-oriented evaluation. The datasets are large and the paper reports not only accuracy metrics but also inference latency, which matters for CI/security scanning pipelines.
* Structured representation is thoughtful. The HRG design (heterogeneous node/edge types plus regex-extended semantic edges) is more faithful to regex semantics than pure token or plain AST baselines.
* Component contributions are partially justified. Ablations (e.g., kernel-guided motif distillation, attention fusion) suggest the proposed pieces do matter.

Weaknesses:
* Core false-positive/false-negative causes remain. Even with better structure modeling, the method still largely learns “structural similarity → vulnerability likelihood.” ReDoS exploitability depends on engine-specific matching behavior and input-dependent backtracking paths; similar-looking structures can behave very differently.
* Lack of dynamic semantics and engine fidelity. ReDoS is fundamentally about the interaction between regex and the actual engine implementation (PCRE vs Java vs Python vs .NET, etc.). A purely static predictor struggles to represent these differences. A more convincing direction would be model-guided dynamic validation (e.g., guided fuzzing/payload search with real engine feedback).
* Limited novelty in the broader ML sense. The contribution reads like applying a fairly standard “GNN on program/structure graphs + handcrafted motifs” recipe to a known security task. The motifs appear partly manual/heuristic, which weakens the methodological novelty claim.
* Potentially weak comparison to stronger learning baselines. The paper emphasizes comparisons to classical tools; however, stronger modern baselines (Transformer/LLM-based encoders, modern graph transformers, or other structure-aware sequence models) are not prominently or convincingly addressed.
* Adoption/maintenance cost questions. Building and maintaining a graph+feature pipeline plus training may be heavier than the direction many practitioners are moving toward (LLM-based scanners, dynamic mitigations, or hybrid approaches).
* Presentation issue. Figure 1(iii) contains severely compressed/distorted text, reducing readability. And one of the references has a misspelled name.

---

> ### Author Rebuttal · Authors · 2026-03-30
>
> We sincerely thank reviewer pJ9L for detailed reading and meaningful feedback.
>
> **W1: Structural Information in ReDoS Detection**
>
> **RW1:** We agree that ReDoS detection is highly engine-dependent. To capture engine-specific matching semantics, our method relies on datasets where ground truth labels were strictly generated and validated using their respective native engines, as described in the engines and ground truth paragraphs of Section~5.1. Because the labels inherently encode the superlinear or linear backtracking behaviors exhibited by each engine, RMGNN learns to map regexes to actual engine-specific vulnerabilities from engine-validated supervision.
>
> Furthermore, since a regex can be viewed as a program for string processing, whose operators capture the rich matching semantics of engines, static approaches like RegexStatic and Rexploiter focus on analyzing the structural properties or automata representations of regexes to detect ReDoS vulnerabilities.
>
> **W2: Dynamic Validation**
>
> **RW2:** We agree that dynamic validation is useful for confirming vulnerability exploitation. However, our paper makes a trade-off by prioritizing scalability over dynamic validation: RMGNN is designed to serve as a scalable detector for large-scale and real-time scenarios, and can also be used in vulnerability exploitation for quick detection. In real-world software supply chains or CI/CD pipelines, executing dynamic payload searches for thousands of regexes introduces unacceptable latency. RMGNN achieves sub-second inference speeds (averaging 0.031s per regex) that is indispensable for proactive, large-scale vulnerability scanning. As suggested, we will add a discussion of this direction (model-guided dynamic validation) in the related and future works of the paper.
>
> **W3 \& Q3: Methodological Novelty \& Handcrafted Motifs**
>
> **RW3 \& RQ3:**  Please refer to our response to W1, raised by Reviewer 6Jg7, for a detailed clarification of the role of motifs and the novelty of our proposed GNN framework. In addition, the proposed HRG preserves the comprehensive heterogeneous semantics of regexes with extended features.
>
> **Q1 \& Q2 \& W3: Stronger Baselines \& LLM Comparisons**
>
> **RQ1 \& RQ2 \& RW3:** To address these concerns, we further expanded the experiments to incorporate recent sequence-based models, transformer-based models, and modern graph transformers. As shown in Table 1, RMGNN consistently achieves the best performance, outperforming the best performing baseline GraphGPS, by 11.68 and 14.30 F1 scores on Corpus and PyPI, respectively. Detailed comparisons with LLM-based methods are provided in our response to Q4 raised by Reviewer ANcd. The complete results and the updated experimental environment will be included in the revised manuscript.
>
> ### Table 1.
> |Method|Corpus||||PyPI||||
> |-|-|-|-|-|-|-|-|-|
> ||TP|Acc|Rec|F1|TP|Acc|Rec|F1
> |SAT|1,534|68.75|37.26|41.94|8,974|75.06|34.29|37.87
> |Transformer|1,363|66.24|33.11|37.26|7,742|72.98|29.58|32.67
> |GAT|1,602|78.28|38.91|52.05|10,734|73.55|41.02|40.73
> |Graphormer|1,874|76.40|45.52|53.88|11,262|72.95|43.03|41.35
> |HGT|2,215|79.06|53.80|60.89|9,502|78.95|36.31|43.33
> |GraphGPS|2,284|82.73|55.48|66.05|12,493|81.02|47.74|52.72
> |**RMGNN**|**2,999**|**87.36**|**72.84**|**77.73**|**17,069**|**85.78**|**65.22**|**67.02**
>
> **W4: Adoption and Maintenance Cost**
>
> **RW4:** We would like to clarify that RMGNN is a lightweight model specialized for ReDoS detection, whose overall adoption and maintenance cost is relatively low, rather than a large general-purpose neural architecture. To make this point clearer, we further report preprocessing and training time, total runtime, and GPU memory usage on the PyPI dataset in Table 2. Even when considering the entire graph-and-feature construction pipeline, training, and inference together,  RMGNN achieves a 51.64× speedup over the fastest conventional baseline (ReDoSHunter) and a 146.84× speedup over the fastest LLM baseline (QwQ-32B-AWQ). Its GPU memory is 93.25\% lower than the average memory usage of the LLM baselines. These results demonstrate that RMGNN not only improves detection performance but also maintains a substantially lower practical cost.
>
> Note: The reported runtime is the sum of the execution time across all threads used in the process.
>
> ### Table 2.
>
> |Method|Preprocessing&Training Time/s|Total Runtime/s|GPU Memory/mib|
> |-|-|-|-|
> |Regulator|-|666078.36|-
> |ReDoSHunter|-|542074.41|-
> |Qwen3-Coder-Next|-|1993723.19|196,512
> |QwQ-32B|-|2745811.97|92,136
> |QwQ-32B-AWQ|-|1541313.67|81,920
> |GPT-OSS-20B|-|3156725.52|40,960
> |**RMGNN (Ours)**|**7426.49**|**10497.06**|**6,940**
>
> **W5: Presentation Issues**
>
> **RW5:** We deeply appreciate your meticulous reading. We will redraw Figure 1(iii) in a vector graphic format and correct the misspelled word "Rcale" to "Scale" in the references in our revised manuscript.
>
> If your concerns have been addressed, could you kindly consider raising your score? We greatly appreciate your comments and support.

---

> > ### Author Rebuttal · Reviewer_pJ9L · 2026-04-03
> >
> > Thanks the authors for their responses.
> >
> > With the additional experiments and clarifications provided in the rebuttal, my earlier concerns have been addressed, particularly those regarding the role of motifs and the comparisons with stronger modern baselines.

---

> > > ### Author Response · Authors · 2026-04-04
> > >
> > > We sincerely thank you for your careful consideration of our rebuttal. We are pleased that our responses have helped clarify and address your concerns. We realize that this may be a somewhat bold request, but if you feel that your concerns have been sufficiently resolved, we would be truly grateful if you would consider updating your overall recommendation rating to a positive one. This would mean a great deal to us. If you have any further questions or concerns, we remain available at any time and would be glad to clarify them.

---

### Official Review · Reviewer_6Jg7 · 2026-03-12

**Soundness:** 3
**Presentation:** 3
**Significance:** 3
**Originality:** 3
**Overall Recommendation:** 5
**Confidence:** 3

**Summary:**

This paper proposes a motif-based method for ReDoS detection. The method first extends ASTs with several attributes and adopts the number of three kinds of motifs as part of node/edge features, resulting in HRGs. Then, RMGNN performs propagation on HRGs and uses teacher-student model distillation to train the prediction model.

**Compliance With Llm Reviewing Policy:**

Affirmed.

**Final Justification:**

The authors have addressed my primary concerns in the rebuttal. Overall, this work presents a solid application of GNNs, and the effectiveness of the proposed method is well-supported.

**Key Questions For Authors:**

1. Table 2 shows that the performance of the model without motif enhancement (i.e., RMGNN (HRG)) drops significantly compared to the full model. However, according to existing literature[R1], GNNs are able to learn the number of subgraphs. Can you explain why RMGNN cannot capture the ReDoS pattern without pre-computed motif numbers? Discussing the capacity of GNNs to learn zero-day ReDoS patterns without explicit human-injected motifs would make the paper much more valuable.
2. Related literature: RegexClassifier [R1] also solves the regex classification problem using graph learning. Though the exact downstream task is different, the input (regex) and output (classification) formats are identical between your method and RegexClassifier. Therefore, I think it could serve as a highly relevant graph-based baseline for comparison.

[R1] "RegexClassifier: A GNN-Based Recognition Method for State-Explosive Regular Expressions," 2023 IEEE Symposium on Computers and Communications (ISCC)

[R2] Counting Graph Substructures with Graph Neural Networks

**Limitations:**

The heavy reliance on injected human knowledge (e.g., explicitly defining counting-centered motifs and rule-based semantic edges) reduces the proposed method to a heuristic engineering effort for ReDoS detection, rather than leveraging the intrinsic learning capabilities of GNNs to solve the problem.

**Strengths And Weaknesses:**

Strengths:

1. Formulating ReDoS detection as a graph learning problem is a novel and promising direction, particularly given the low-latency requirements in practical deployment scenarios.
2. The proposed method is clearly presented and its effectiveness is well-demonstrated. The inference efficiency improvement (e.g., up to a 244× speedup) is highly significant compared to existing dynamic or hybrid testing methods.

Weaknesses:

1. The most critical knowledge for predicting ReDoS in the proposed framework relies heavily on hand-crafted features rather than automated representation learning. The pre-computed motif occurrences and the WL-kernel for fine-grained local structural similarity within k hops are all pre-defined domain priors. This makes the proposed method function more like a heuristic-driven approach augmented by a neural classifier, which diminishes the technical novelty of the graph neural network itself.
2. The hyperparameter sensitivity analysis reveals the fragile robustness of the trained model. As shown in Figure 5, the model suffers from severe performance degradation (over-smoothing) when the propagation layers exceed 3, and easily overfits when the hidden dimension increases from 128 to 256. I think further exploration is needed for better generalization.

---

> ### Author Rebuttal · Authors · 2026-03-30
>
> We sincerely appreciate Reviewer 6Jg7's thoughtful comments.
>
> **W1: Hand-Crafted Priors and Novelty of the GNN Framework**
>
> **RW1:** These motif priors play an auxiliary role in the representation learning process rather than dominating it. In addition, the WL-kernel serves as a training-time structural regularizer through the distillation loss. Overall, the framework is fundamentally driven by automated representation learning, as supported by the following points.
>
> **1. Auxiliary role of motifs.** The motifs are deliberately designed at a abstract level, and they do not encode fine-grained patterns. Relying solely on these motif occurrences without GNN-based learning would inevitably trigger many false positives, thus they are not sufficient by themselves for ReDoS prediction. When enhanced with these motifs, our model achieves the best performance, as shown in Table~1 of our paper.
>
> **2. Core role and optimization of the GNN framework.** We specifically design RMGNN with Heterogeneous Information Propagation, Kernel-Guided Motif Structure Learning, and Cross-Attention Global-Local Fusion to explicitly tailor GNNs for the ReDoS learning domain. As shown in Table~2 of our paper, even without motif enhancement, RMGNN still outperforms a standard GCN under the same HRG representation. For example, on Corpus, RMGNN (HRG) achieves 65.04\% F1, outperforming GCN (HRG) at 53.35\% by 11.69\%. The same trend is also observed on PyPI (42.20\% vs. 35.53\%), and Maven (34.79\% vs. 33.31\%). This suggests that the performance gains are attributable to the proposed GNN design itself.
>
>
> **W2: Hyperparameter Sensitivity and Model Robustness**
>
> **RW2:** We sincerely thank the reviewer for this helpful suggestion. We agree that GNN always suffers from over-smoothing. To mitigate the limitation, we design a residual connection in the final fusion module to preserve gradient flow and improve training stability. Although this is not a dedicated mechanism for eliminating degradation under all hyperparameter settings, it helps stabilize the selected configuration that is analyzed by the hyperparameter experiment (Fig.~5) in practice. We agree that broader generalization under wider hyperparameter ranges deserves further study in this domain.
>
>
> **Q1: Whether GNNs Can Learn ReDoS-Relevant Substructures without Explicit Motif Enhancement**
>
> **RQ1:** We agree that prior work such as [R2] has shown that GNNs can, in principle, learn subgraph-counting information. Consistent with this observation, Table 1 demonstrates that RMGNN retains substantial learning capacity even without motif enhancement: under the same HRG representation, it consistently outperforms a standard GCN. Therefore, the motifs are introduced to strengthen automated representation learning rather than to replace it.
>
> To further address the reviewer’s concern regarding zero-day generalization in the absence of the motifs, we additionally compare RMGNN (HRG) with the full model, RMGNN (MEHRG), on 340 real-world zero-day ReDoS cases collected from JavaScript CVEs. As shown below, even without motif enhancement, RMGNN (HRG) already achieves strong zero-day performance. Compared with the full model, the performance drop is moderate: TP decreases by 18, accuracy by 2.40 points, recall by 5.35 points, and F1 by 3.07 points. These results suggest that zero-day generalization does not rely solely on motif enhancement.
>
> ***Table 1***
>
> | Method | TP | Acc | Rec | F1 |
> |---|---:|---:|---:|---:|
> | RMGNN (HRG) | 283 | 92.95 | 84.23 | 91.44 |
> | RMGNN (MEHRG) | 301 | 95.35 | 89.58 | 94.51 |
>
> **Q2: Comparison with RegexClassifier [R1]**
>
> **RQ2**: We sincerely thank the reviewer for highlighting [R1]. We agree it is relevant and strongly validates our motivation to apply GNNs to ReDoS tasks. [R1] proposes a GNN model that takes NFAs as input to predict DFA state exploration, thus it is a compatibility challenge to apply its architecture to ReDoS task. The main reason is as follows:
>
> **Limitation of Supporting Extended Features in NFA.** [R1] converts regexes into NFAs and proposes a GNN model that takes NFAs as input to predict DFA state exploration. However, the real-world regexes considered in our work are driven by extended features (e.g., lookarounds, backreferences). These features can make the regexes non-regular, and thus it is computationally impossible to represent them using NFAs. Consequently, directly applying the architecture of [R1] to the ReDoS detection task becomes challenging. As suggested, we will explicitly discuss [R1] in the "Related Works" section to broaden our literature review and reinforce our motivation to apply GNNs to ReDoS tasks.
>
> If your concerns have been addressed, could you kindly raise the score? We greatly appreciate your comments and support.

---

> > ### Author Rebuttal · Reviewer_6Jg7 · 2026-04-04
> >
> > Thank you for the rebuttal. My main concerns are resolved and I will raise my score.

---

> > > ### Author Response · Authors · 2026-04-04
> > >
> > > We sincerely appreciate your careful consideration of our rebuttal and your decision to raise your score.  We are pleased that our responses helped address your concerns and promote a clearer mutual understanding.

---

### Decision · Program_Chairs · 2026-04-30

**Decision:**

Accept (regular)

**Comment:**

This paper proposes a graph learning solution to detecting 'poisonous' regular expressions.  Reviewers praised the novelty of the application, and agreed that the parts of the solution work together to further the result.  However there is a sense that this is primarily an application paper, and doesn't add much novelty in the broader ICML sense.  The strongest argument in the negative direction is that the framework does rely heavily on hand-craft featurization, which might limit its generality to new domains.

Strengths:
1. Problem formulation (ReDoS detection as a graph learning problem) is novel
2. Ablations (e.g., kernel-guided motif distillation, attention fusion) suggest the proposed pieces do matter.
3. method shows strong performance in benchmark datasets.

Weakness:
1. Limited novelty in the broader ML sense.
2. The most critical knowledge for predicting ReDoS in the proposed framework relies heavily on hand-crafted features
3. The approach detects vulnerability but does not generate attack strings or quantify severity.